Brief Communication

# Structural basis of LRPPRC–SLIRP-dependent translation by the mitoribosome

Vivek Singh[1,8], J. Conor Moran[2,8], Yuzuru Itoh[1,6] ✉, Iliana C. Soto[3], Flavia Fontanesi[2], Mary Couvillion[3], Martijn A. Huynen[4], L. Stirling Churchman[3] ✉, Antoni Barrientos[5] ✉ & Alexey Amunts [1,7] ✉

In mammalian mitochondria, mRNAs are cotranscriptionally stabilized by the protein factor LRPPRC (leucine-rich pentatricopeptide repeat-containing protein). Here, we characterize LRPPRC as an mRNA delivery factor and report its cryo-electron microscopy structure in complex with SLIRP (SRA stem-loop-interacting RNA-binding protein), mRNA and the mitoribosome. The structure shows that LRPPRC associates with the mitoribosomal proteins mS39 and the N terminus of mS31 through recognition of the LRPPRC helical repeats. Together, the proteins form a corridor for handoff of the mRNA. The mRNA is directly bound to SLIRP, which also has a stabilizing function for LRPPRC. To delineate the effect of LRPPRC on individual mitochondrial transcripts, we used RNA sequencing, metabolic labeling and mitoribosome profiling, which showed a transcript-specific influence on mRNA translation efficiency, with cytochrome c oxidase subunit 1 and 2 translation being the most affected. Our data suggest that LRPPRC–SLIRP acts in recruitment of mitochondrial mRNAs to modulate their translation. Collectively, the data define LRPPRC–SLIRP as a regulator of the mitochondrial gene expression system.

The mitoribosome consists of at least 82 proteins and three rRNAs with 13 modified nucleotides[1,2]. The mitoribosome is organized into a small subunit (SSU) and large subunit (LSU) that are assembled from multiple components in a coordinated manner and through regulated sequential mechanisms[3–8]. SSU formation is accomplished by the association of the mitoribosomal protein mS37 and the initiation factor mtIF3, leading to a mature state that is ready for translation of the mRNA[6,9]. In mammals, mitochondrial transcription is polycistronic and gives rise to two long transcripts, corresponding to almost the entire heavy and light mitochondrial DNA (mtDNA) strands. The individual mRNAs are available for translation only after they are liberated from the original polycistronic transcripts and polyadenylated[10]. In *Escherichia coli*, a functional transcription–translation coupling mechanism has been characterized involving a physical association of the RNA polymerase with the SSU, termed the expressome[11–13]. In contrast, in mammalian mitochondria, nucleoids are not compartmented with protein synthesis; mitoribosomes are independently tethered to the membrane[14,15] and no coupling with the RNA polymerase has been reported. In addition, human mitochondrial mRNAs and the mitoribosome do not have the Shine–Dalgarno (SD) and anti-SD sequences that are used in bacteria to recruit mRNAs[16]. Mitochondrial mRNAs also lack cap 5′ modifications, which are a hallmark of eukaryotic cytosolic translation initiation.

[1]Science for Life Laboratory, Department of Biochemistry and Biophysics, Stockholm University, Solna, Sweden. [2]Medical Scientist Training Program, Department of Biochemistry and Molecular Biology, University of Miami Miller School of Medicine, Miami, FL, USA. [3]Blavatnik Institute, Department of Genetics, Harvard Medical School, Boston, MA, USA. [4]Department of Medical BioSciences, Radboud University Medical Center, Nijmegen, The Netherlands. [5]Department of Neurology, University of Miami Miller School of Medicine, Miami, FL, USA. [6]Present address: Department of Biological Sciences, Graduate School of Science, University of Tokyo, Tokyo, Japan. [7]Present address: Westlake University, Hangzhou, China. [8]These authors contributed equally: Vivek Singh, J. Conor Moran. ✉e-mail: yuzuru.itoh@bs.s.u-tokyo.ac.jp; churchman@genetics.med.harvard.edu; abarrientos@med.miami.edu; alexey.amunts@gmail.com

In the cytosol, mRNA is recruited to a preinitiation complex, consisting of the SSU and translation initiation factors, which then scans along the 5′ untranslated region to find the start codon[17,18]. No equivalent mechanism has been found in mitochondria; thus, how mRNAs are delivered for translation in mitochondria has remained unknown.

The 130-kDa protein factor LRPPRC (leucine-rich pentatricopeptide repeat-containing protein), a member of a Metazoa-specific pentatricopeptide repeat family, was reported to act as a global mitochondrial mRNA chaperone that binds cotranscriptionally[19–21]. LRPPRC is an integral part of the post-transcriptional processing machinery required for mRNA stability, polyadenylation and translation[19–22]. Mutations in the gene encoding LRPPRC lead to Leigh syndrome, French-Canadian type (LSFC), an untreatable pediatric neurodegenerative disorder caused by ultimately impaired mitochondrial energy conversion[23].

LRPPRC has been reported to interact with a small 11-kDa protein cofactor SLIRP (SRA stem-loop-interacting RNA-binding protein)[22,24] that has roles in LRPPRC stability and the maintenance of steady-state mRNA levels[25]. *SLIRP* silencing results in the destabilization of respiratory complexes, loss of enzymatic activity and a reduction in mRNA levels, implicating a role in mRNA homeostasis[26]. *SLIRP* variants cause a respiratory deficiency that leads to mitochondrial encephalomyopathy[27]. In addition, *SLIRP* knockdown results in increased turnover of LRPPRC[25,27,28] and in vivo costabilization suggests that the two entities have interdependent functions[25,29]. The interaction of LRPPRC and SLIRP in vitro has been previously studied[30].

LRPPRC has also been implicated in coordinating mitochondrial translation[21,31]. Previous analysis showed a correlation between the presence of LRPPRC and mRNA on the mitoribosome[32]. However, there are no structures available for LRPPRC, SLIRP or any complexes containing them and in vitro reconstitution could not provide meaningful information, in part because not all the components of the mitochondrial gene expression system have been characterized. Thus, although isolated mitoribosomal models have been determined[2,33,34], the molecular mechanisms of mRNA delivery to the SSU for activation of translation and the potential involvement of LRPPRC–SLIRP in this process have remained unknown.

## Results
### Structure determination of LRPPRC–SLIRP with the mitoribosome

To explore the molecular basis for translation activation in human mitochondria, we used low-salt conditions to isolate a mitoribosome in complex with LRPPRC–SLIRP–mRNA for cryo-electron microscopy (cryo-EM). We merged particles containing transfer RNA (tRNA) in the P-site, as well as a region with extra density in the vicinity of the mRNA entry channel and applied iterative local-masked refinement and classification with signal subtraction (Extended Data Fig. 1a). This resulted in a 2.9-Å resolution map of the mitoribosome during mRNA delivery to the SSU, with local resolution for the LRPPRC-binding region of ~3.4 Å (Table 1, Extended Data Fig. 1b,c). The reconstruction showed a clear density only for the LRPPRC N-terminal domains (residues 64–644; average local resolution, ~4.5 Å) bound to the SSU head, which is consistent with a previous mass-spectrometry analysis (Extended Data Fig. 1d,e)[32]. This allowed us to model 34 α-helices, 17 of which (α2–α18) form a ring-like architecture, while the remainder form an extended tail that adopts a 90° curvature and projects 110 Å from the SSU body in parallel to the L7–L12 stalk (Fig. 1a,b). The C-terminal domains (residues 645–1394) were not resolved. The complete LRPPRC model obtained with AlphaFold2 (ref. 35) combined with translation, liberation and screw-motion determination (TLSMD) analysis[36,37] defined the C-terminal domains as individual segments, indicating potential flexibility (Extended Data Fig. 2).

## Table 1 | Cryo-EM data collection, refinement and validation statistics

| | Monosome with LRPPRC–SLIRP (PDB 8ANY), (EMD-15544) |
|---|---|
| **Data collection and processing** | |
| Electron microscope | Titan Krios |
| Camera | K2 summit |
| Magnification | ×165,000 |
| Voltage (kV) | 300 |
| Electron exposure (e⁻ per Å²) | 29–32 |
| No. of frames | 20 |
| Defocus range (μm) | −0.6 to −2.8 |
| Pixel size (Å) | 0.83 |
| Symmetry imposed | $C_1$ |
| Final particle images (no.) | 41,815 |
| Map resolution (Å) (overall/LSU body/CP/L10–L12 stalk/SSU body/SSU head/SSU tail/mS39–LRPPRC–SLIRP) | 2.85/2.69/3.07/3.07/2.89/2.84/3.02/3.37 |
| FSC threshold | 0.143 |
| Map resolution range (Å) | 2.5–8.0 |
| **Refinement** | |
| Initial model used (PDB code) | 6ZSG, 6RW4 |
| Model resolution (Å) | 2.7 |
| Model to map CC (CC$_{volume}$) | 0.84 |
| FSC threshold | 0.5 |
| Map sharpening $B$ factor (Å²) (overall/LSU body/CP/L10–L12 stalk/SSU body/SSU head/SSU tail/mS39–LRPPRC–SLIRP) | −33/−28/−58/−49/−39/−39/−55/−60 |
| Model composition | |
| Nonhydrogen atoms | 356,138 |
| Hydrogen atoms | 160,529 |
| Protein chains | 90 |
| RNA chains | 7 |
| Protein residues (nonmodified/*N*-acetylalanine/*N*-acetylserine/*N*-acetylthreonine/O¹-methylisoaspartate) | 15,624/3/1/1/1 |
| RNA residues (nonmodified/mG/mU/m¹A/m²G/ψ/m⁴C/m⁵C/m⁵U/m⁶₂A) | 2,822/2/1/2/1/2 |
| Ligands (ATP/GDP/NAD/2Fe–2S/spermine/spermidine/putrescine) | 1/1/1/3/1/4/1 |
| Ions (Zn²⁺/K⁺/Mg²⁺) | 3/49/206 |
| Waters | 6,926 |
| **Validation** | |
| Ramachandran plot | |
| Favored (%) | 98.11 |
| Allowed (%) | 1.85 |
| Disallowed (%) | 0.05 |
| Clashscore | 2.64 |
| Poor rotamers (%) | 0.00 |
| Root-mean-square deviations | |
| Bonds (Å) | 0.002 |
| Angles (°) | 0.432 |
| C$_\beta$ outliers (%) | 0.00 |
| CaBLAM outliers (%) | 0.9 |

PDB, Protein Data Bank; GDP, guanosine diphosphate.

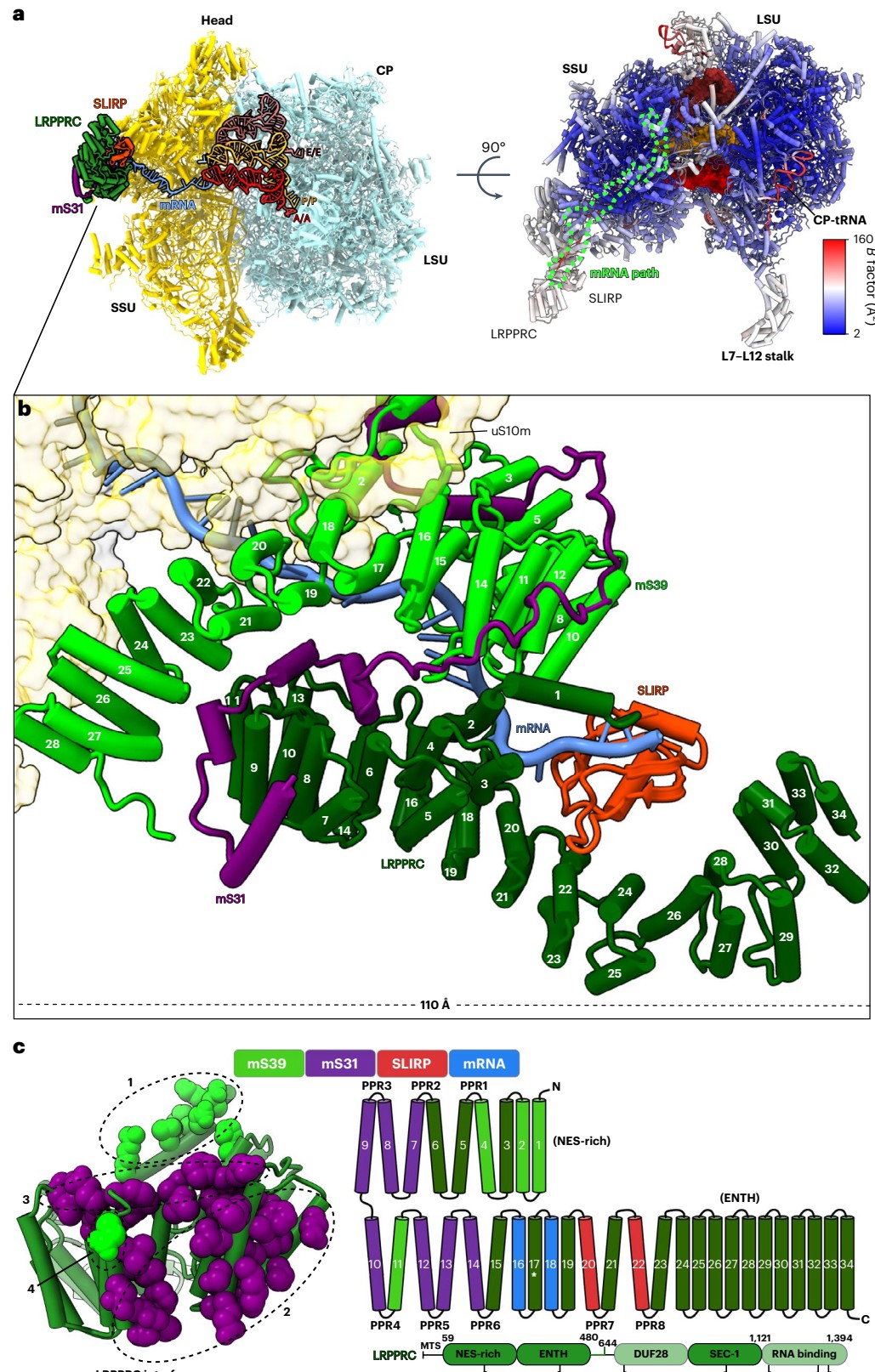

**Fig. 1 | Structure of mitoribosome with LRPPRC–SLIRP bound to mRNA.**
**a**, Model overview of the mitoribosome in complex with LRPPRC–SLIRP. Right, top view of the model colored by atomic *B* factor (Å²). tRNAs are shown in surface representation (red, orange and brown). The mRNA path (light green) is highlighted. **b**, A close-up view of the interactions within the mitoribosome in complex with LRPPRC–SLIRP–mRNA. LRPPRC associates with mS31–m39 through a ring-like structure (α2–α18), together forming a corridor for the handoff of the mRNA from SLIRP. **c**, Contact sites between LRPPRC and mS31–mS39 (within 4-Å distance); view from the interface. Right, schematic diagram showing the topology of LRPPRC consisting of 34 helices. Colors represent engagement in interactions with mS39 (light green), mS31 (purple), SLIRP (orange) and mRNA (blue). The position of the LSFC variant (A354V) is indicated with an asterisk on α17.

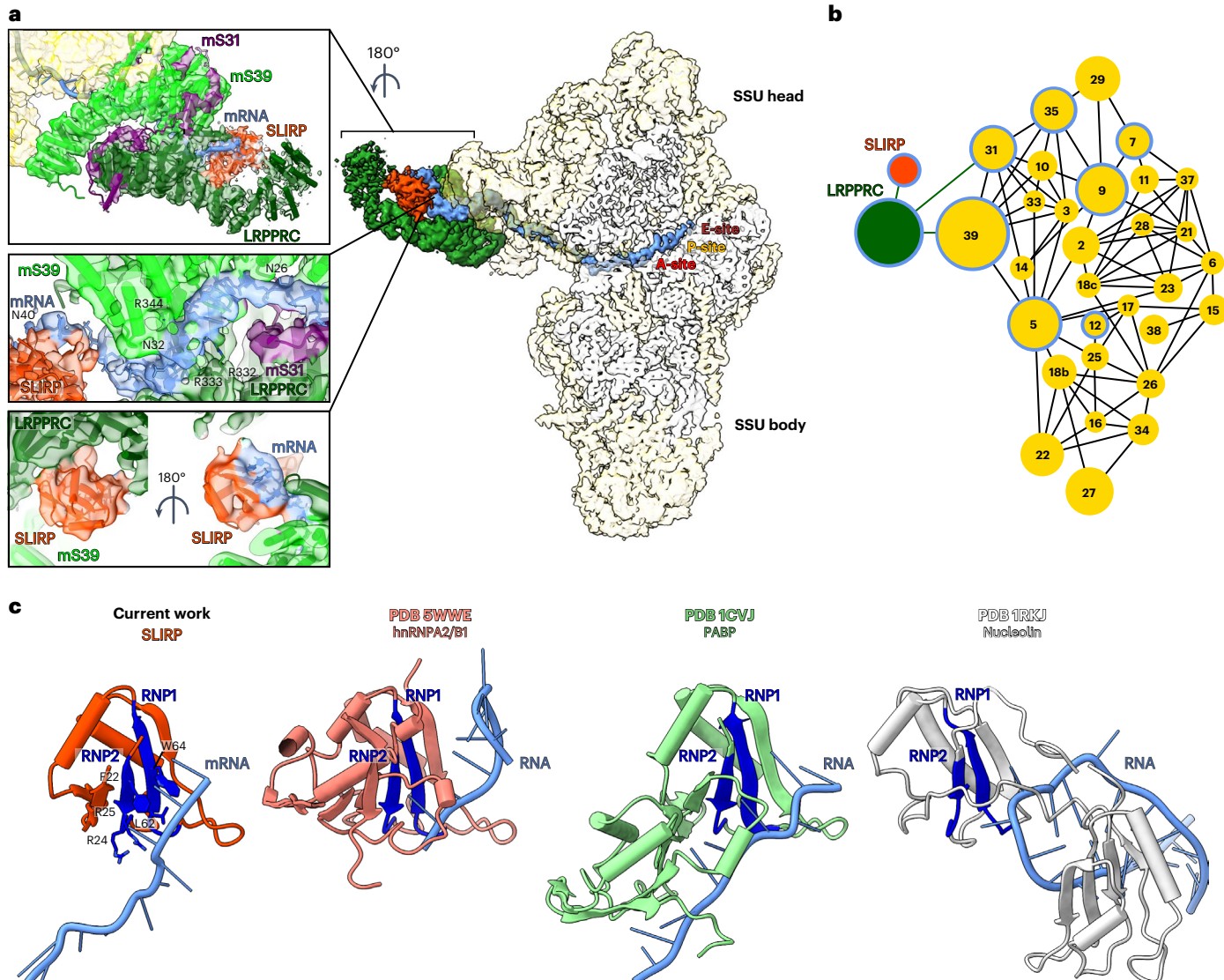

**Fig. 2 | Overview of density for LRPPRC, SLIRP and mRNA and their interactions with SSU proteins. a,** The density map for LRPPRC (dark green), SLIRP (orange) and mRNA (blue) on the SSU is shown in the center. Left, the model and map for mS39–LRPPRC–SLIRP and corresponding bound mRNA residues in close-up views. Arginine residues involved in mRNA binding are indicated. Bottom, close-up views of SLIRP with its associated densities for LRPPRC and mRNA. For clarity, the map for SLIRP was low-pass filtered to 6-Å resolution. **b,** Schematic of protein–protein interactions, where node size corresponds to relative molecular mass. Nodes of proteins involved in mRNA binding are encircled in blue. **c,** RRM-containing proteins (SLIRP, heterogeneous nuclear ribonucleoproteins A1/B2 (hnRNPA1/B2; PDB 5WWE), poly(A)-binding protein (PABP; PDB 1CVJ) and nucleolin (PDB 1RKJ)) are shown in complex with RNA, with RNP1 and RNP2 submotifs colored blue.

When LRPPRC–SLIRP is bound to the mitoribosome, a previously disordered density of mS31 that extends from the core also becomes ordered, revealing its N-terminal region (Fig. 2a). This region is arranged in two helix–turn–helix motifs, offering a surface area of 1,930 Å² for direct interactions with LRPPRC (Figs. 1c and 2). The position of LRPPRC residue 354, at which the substitution A354V leads to LSFC with a clinically distinct cytochrome c oxidase deficiency and acute fatal acidotic crises, is in a buried area of α17, close to the mRNA-binding region (Fig. 1c and Extended Data Fig. 2a). A previous study demonstrated that this substitution abolishes the interaction with the protein SLIRP[38]. Consistent with mass-spectrometry analysis[30] and the interaction interface previously determined[38], the remaining associated density was assigned as SLIRP, found to be located close to the Epsin N-terminal homology (ENTH) domain of LRPPRC (Fig. 2a). Lastly, SLIRP is connected to an elongated density on the LRPPRC surface that is also associated with six of the mitoribosomal proteins and corresponds to the endogenous mRNA (Fig. 2).

**SLIRP is stably associated with mRNA and LRPPRC on the SSU**

The binding of SLIRP in our model is enabled by LRPPRC helices α20 and 22, which is consistent with cross-linking mass-spectrometry data and mutational analysis[30]. The structure reveals that SLIRP links the nuclear export signal (NES) domain with the curved region of the ENTH domain of LRPPRC (Fig. 1b,c). This binding of SLIRP contributes to a corridor for the mRNA that extends to mS31 and mS39 (Fig. 1b and Supplementary Video 1). Through this corridor, the mRNA extends over ~180 Å all the way to the decoding center (Fig. 2a). In our structure, SLIRP is oriented such that the conserved RNA recognition motif (RRM), including its submotifs RNP1 (residues 21–26) and RNP2 (residues 60–67)[39–41], form an interface with modeled mRNA (Fig. 2). The arrangement of RNP1 and RNP2 with respect to the mRNA is similar to that observed in previously reported structures of other RRM proteins[42–44] (Fig. 2c and Extended Data Fig. 3). Moreover, residues R24 and R25 of the RNP2 motif and L62 of the RNP1 motif, previously implicated to be required for RNA binding by SLIRP[38], are positioned within an interacting distance of

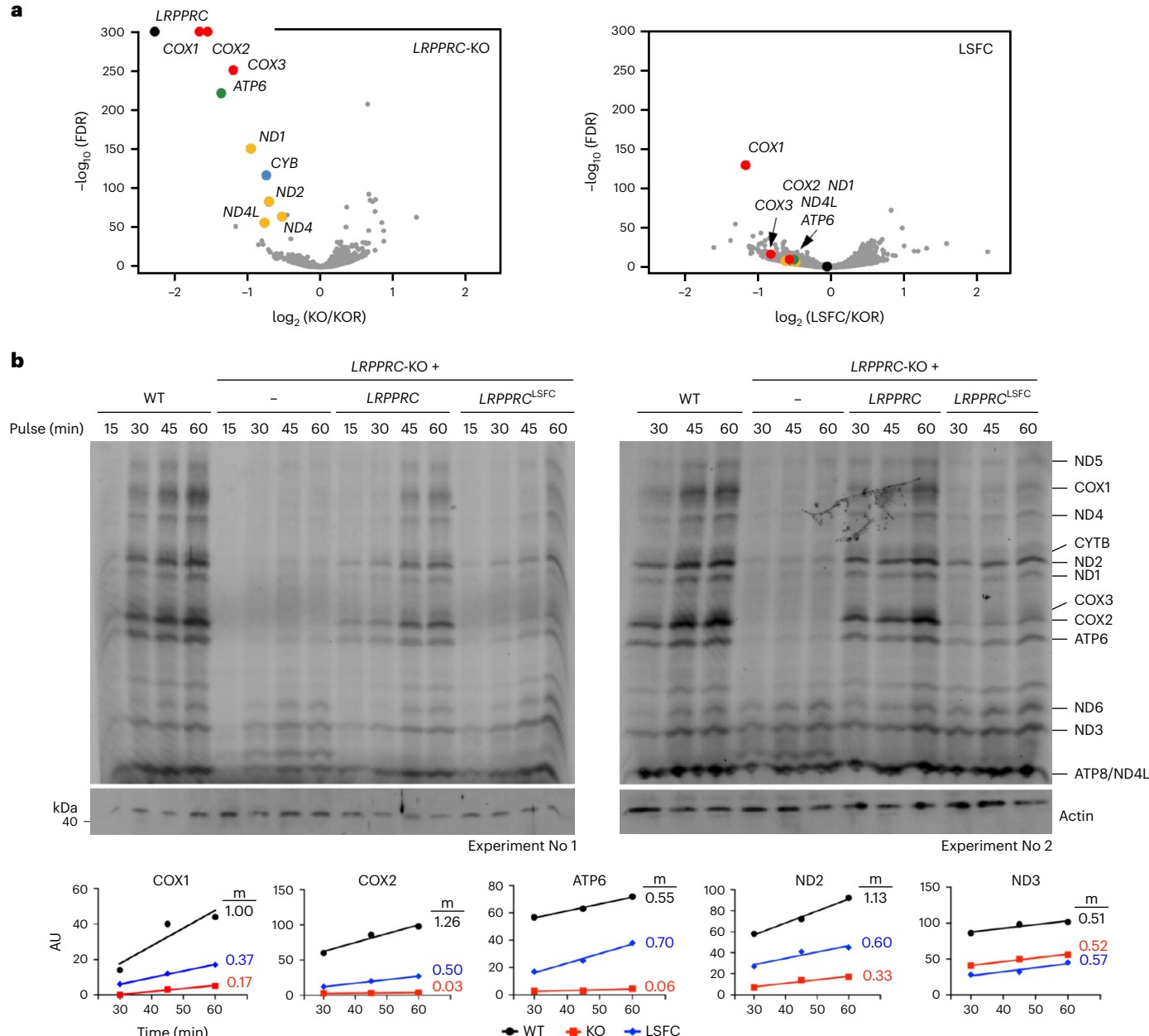

**Fig. 3 | Mechanism of LRPPRC–SLIRP-mediated mRNA binding and stabilization. a**, Whole-cell RNAseq normalized by read depth, comparing *LRPPRC*-KO cells to KO cells reconstituted with WT *LRPPRC* (KOR) or the LSFC variant (A354V). The results are the average of two biological replicates. The differentially expressed mitochondrial transcripts are color-coded: coding for subunits of cytochrome c oxidase in red, NADH dehydrogenase in yellow, coenzyme Q–cytochrome c oxidoreductase in blue and ATP synthase (CV) in green. **b**, Metabolic labeling with [35S]methionine of newly synthesized mitochondrial polypeptides for the indicated times, in the presence of emetine to inhibit cytosolic protein synthesis, in whole HEK293T WT, *LRPPRC*-KO cells and KO cells reconstituted with LRPPRC (KO + WT) or the LSFC variant (KO + LSFC). Bottom, representative plots of [35S]methionine incorporation into specific polypeptides in WT or *LRPPRC*-KO cells. AU, arbitrary units. The images were quantified in two independent experiments.

the mRNA (Fig. 2a). Thus, SLIRP contributes to the LRPPRC-specific scaffold and accounts for a role in binding the mRNA.

The *B* factor distribution of SLIRP in our model is similar to that of LRPPRC, while still lower than some of the more mobile components of the mitoribosome, such as the acceptor arm of the central protuberance (CP)–tRNA[Val] (Fig. 1a). This indicates a functionally relevant association with LRPPRC in terms of stability of binding. Our finding that SLIRP is involved in handoff of the mRNA to the mitoribosome provides a mechanistic explanation for the previous results from biochemical studies showing that SLIRP affects LRPPRC properties in vitro[29,30] and the presentation of the mRNA to the mitoribosome in vivo[25].

Because the expressome-mediating protein NusG was proposed to regulate mRNA unwinding[11] and SSU proteins uS3 and uS4 have an intrinsic RNA helicase activity in *E. coli*[45], we searched for known helicase signature motifs[46] in the LRPPRC sequence but no such motifs were found. In the mitoribosome, where the mRNA channel entry site is located, a bacterium-like ring-shaped entrance is missing, the entrance itself is shifted and its diameter is expanded[2]. The mRNA extends all the way into the head or beak of the SSU stabilized by mitoribosome-specific components: mS39 helical repeats, the mS35 N terminus that extends from the side of the SSU head and an N-terminal extension of uS9m that contacts the mRNA nucleotide at position 15 (numbered from the E-site).

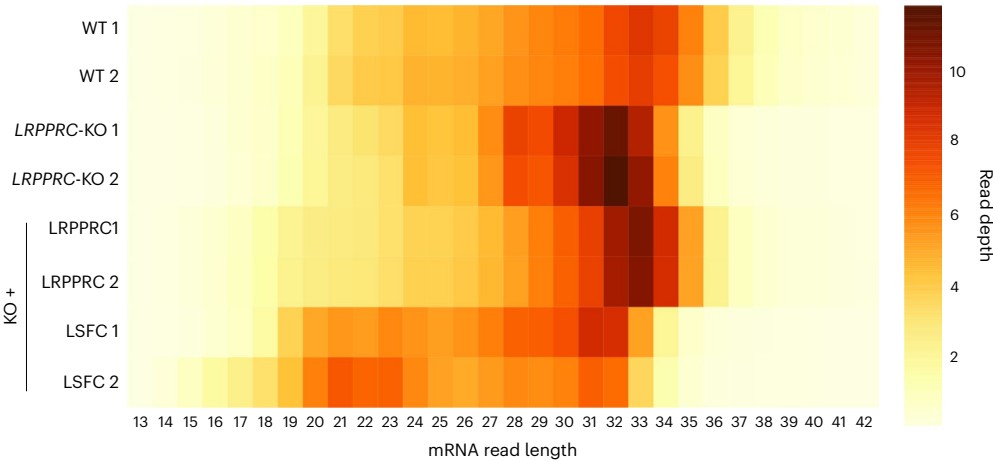

**Fig. 4 | The average length of protected mRNA is decreased in the absence of WT *LRPPRC*.** Heat map showing the length distribution for reads mapping to mitochondrial mRNAs[31] in duplicates of whole HEK293T WT, *LRPPRC*-KO and KO cells reconstituted with WT LRPPRC or its LSFC variant.

## LRPPRC–SLIRP hands off the mRNA to mS31–mS39

Next, we analyzed the structural basis for the complex formation. The association of LRPPRC with the mitoribosome involves helices α1, α2 and α6–α11, which form a mitoribosome-binding surface (Fig. 1b,c). The binding is mediated by four distinct contacting regions (Extended Data Fig. 4): (1) α1–α2 (residues 64–95) is flanked by a region of mS39, a PPR (pentatricopeptide repeat) domain-containing protein, that consists of four bundled helices (α11–α14); (2) α7 and α9 form a shared bundle with two N-terminal helices of mS31 (residues 175–208, stabilized by the C-terminal region of mS39) that encircle the NES-rich domain to complement the PPR domain; (3) α10–α11 are capped by a pronounced turn of mS31 (residues 209–232) acting as a lid that marks the LRPPRC boundary and it is sandwiched by the mS39 helix α19 and C terminus from the opposite side; and (4) α11 is also positioned directly against helix α23 of mS39. Thus, LRPPRC docks onto the surface of the mitoribosome through mS31–mS39, which are tightly associated with each other, and each provides two contact patches to contribute to stable binding.

On the basis of the structural analysis, the handoff of the mRNA for translation is mediated by four of the LRPPRC helices: α1, α2, α16 and α18 (Fig. 1b,c and Supplementary Video 1). The mRNA nucleotides 33–35 (numbered from the E-site) are stabilized in a cleft formed by α1–α2 on one side and α16 and α18 on the other. Nucleotides 31 and 32 contact residues R332 and R333 from LRPPRC and R344 from mS39 (Fig. 2a). This region is within 120–130 Å from the P-site. The involvement of the NES-rich domain of LRPPRC in mRNA binding in our structure is consistent with a biochemical analysis of recombinant LRPPRC where the N-terminal PPR segments were systematically removed, which showed a reduced formation of protein–RNA complexes[30]. The remainder of the mRNA is situated too far from LRPPRC to interact with it. Here, the mRNA is handed to mS31–mS39, consistent with a translation initiation complex[47].

In the structure, mS31, mS39 and LRPPRC together form a 60-Å-long corridor that channels the mRNA from SLIRP toward the mitoribosomal core (Fig. 1b and Supplementary Video 1). With respect to mRNA binding, nucleotides 26–30 bind mS39 PPR domain 5 and nucleotide 26 connects to contacting region 2 (Fig. 2a and Extended Data Fig. 4). Thus, the mRNA handoff is achieved through functional cooperation between LRPPRC–SLIRP and mS31–mS39. Therefore, in the model of the mitoribosome in complex with LRPPRC–SLIRP–mRNA, LRPPRC performs three functions: (1) coordination of SLIRP, which has a key role in the process of mRNA recruitment; (2) association with the SSU; and (3) handoff of the mRNA for translation (Fig. 1, Extended Data Fig. 4 and Supplementary Video 1).

## LRPPRC is recruited for translation of mRNAs

Next, we asked whether LRPPRC–SLIRP delivers all mRNAs to the mitoribosome or is selective. We generated an *LRPPRC*-knockout (KO) cell line[31] that was rescued with either a wild-type (WT) *LRPPRC* or a variant carrying the LSFC founder substitution A354V (ref. 23) (Extended Data Fig. 5). The steady-state levels of the LSFC variant were reduced by 60%, suggesting protein instability as reported in persons with LSFC[22] and the levels of SLIRP were equally decreased (Extended Data Fig. 5). We then implemented an RNA sequencing (RNAseq) approach that confirmed a substantially depleted mitochondrial transcriptome[19,20,48] (Fig. 3a). In the *LRPPRC*-KO cell line, transcripts from the heavy strand were lowered by 1.5–4-fold, except for reduced nicotinamide adenine dinucleotide (NADH) dehydrogenase subunit 3 (ND3), which remained stable, consistent with protein synthesis data (Fig. 3b). The single-light-strand-encoded *ND6* mRNA was not affected as reported[19] and the effect of the LSFC substitution on RNA stability was limited primarily to the cytochrome c oxidase subunit 1 (COX1) transcript (Fig. 3a). Thus, in the *LRPPRC*-KO cell line, all but one of the transcripts from the heavy strand were lowered by 1.5–4-fold, suggesting that LRPPRC's role in heavy-strand mRNA stability is nonspecific.

Metabolic labeling assays using [35S]methionine indicated that incorporation of the radiolabeled amino acid into most newly synthesized mitochondrial proteins was severely decreased in *LRPPRC*-KO cells (Fig. 3b). However, there were differential effects among transcripts; synthesis of ND3, ND4L and adenosine triphosphate (ATP) synthase subunit 8 (ATP8) remained above 50% of the WT but translation of other transcripts proceeded at a lower rate (for example, ND1, ND2 or ATP6) or was virtually blocked (for example, COX1, COX2 or COX3) (Fig. 3b). The translational defect resulted in a decrease in the steady-state levels of the four oxidative phosphorylation (OXPHOS) complexes that contain mtDNA-encoded subunits (Extended Data Fig. 6).

To dissect the role of LRPPRC in translation versus the RNA stability of each transcript, we performed mitoribosome profiling (mitoRPF) along with matched RNAseq (Extended Data Fig. 7). Consistent with the metabolic labeling assays and RNAseq data presented above, we observed a decrease in inferred protein synthesis and RNA abundance for all mitochondrial transcripts except for *ND6* in *LRPPRC*-KO cells (Extended Data Fig. 7a). The visible correlation between synthesis change and RNA change suggests that much of the effect was a result of the changes in RNA abundance alone. Indeed, when synthesis of each transcript was normalized by its abundance to isolate the effect of translation alone (translation efficiency, TE)[49,50], we saw somewhat less dramatic changes at the translation level (Extended Data Fig. 7b).

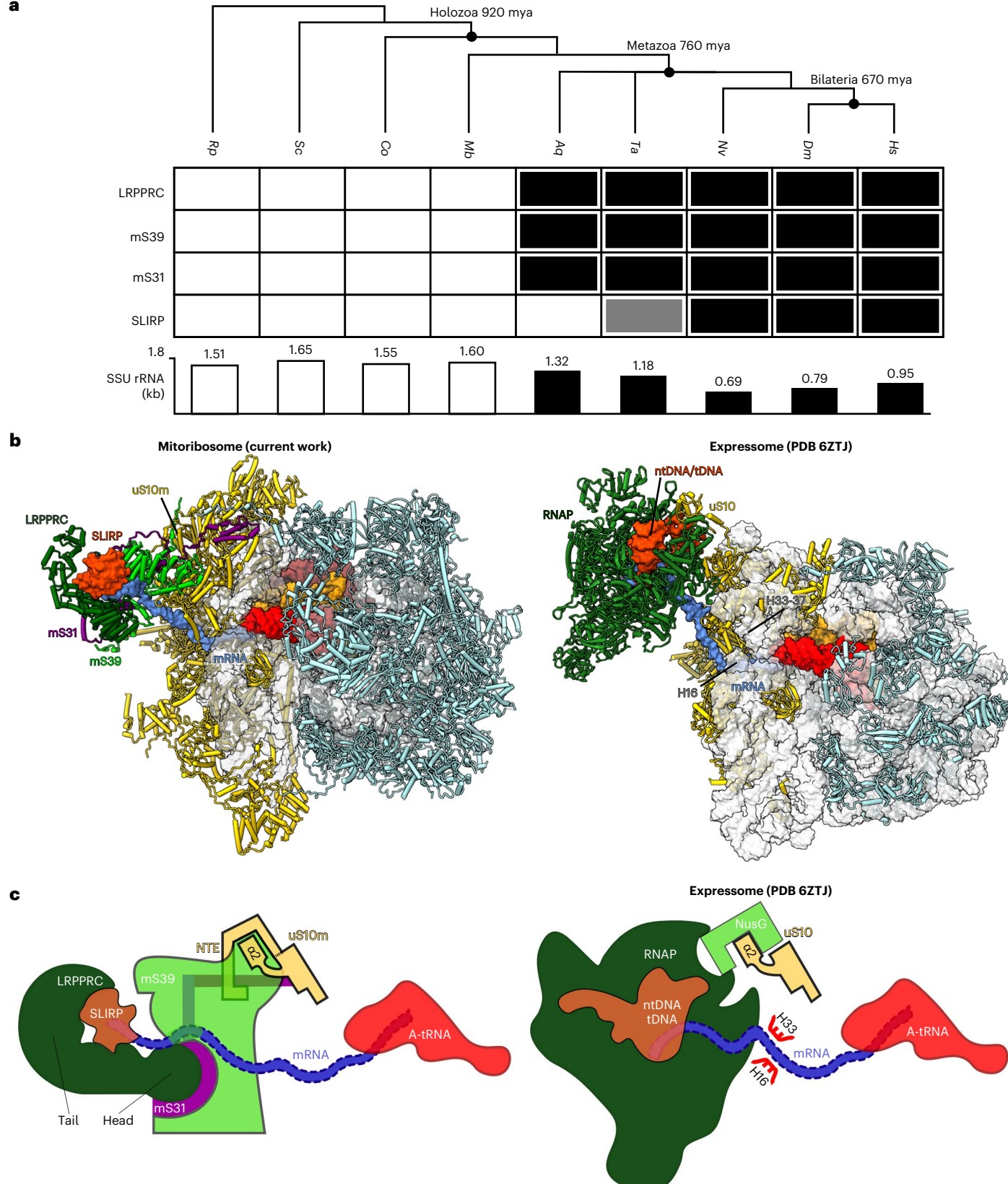

**Fig. 5 | Formation of the mitoribosome in complex with LRPPRC–SLIRP and 70S RNA in complex with RNAP. a**, Phylogenetic analysis showing the correlation between the acquisition of LRPPRC, SLIRP, mS31 and mS39 and the reduction in rRNA in Metazoa. Black rectangles indicate the presence of proteins; gray indicates uncertainty about the presence of an ortholog. *Hs, Homo sapiens; Ds, Drosophila melanogaster; Nv, Nematostella vectensis; Ta, Trichoplax adhaerens; Aq, Amphimedon queenslandica* (sponge);

*Mb, Monosiga brevicollis* (unicellular choanoflagellate); *Co, Capsaspora owczarzaki* (protist); *Sc, Saccharomyces cerevisiae* (fungus); *Rp, Rickettsia prowazekii* (alphaproteobacterium). Dating in million years ago (mya) is based on ref. 60. **b**, Model of the mitoribosome in complex with LRPPRC–SLIRP compared to the uncoupled model of the expressome from *E. coli*[11]. **c**, Schematic representation indicating the association of mRNA-delivering proteins in the mitoribosome compared to the NusG-coupled expressome[11].

This analysis highlights the differential effects across transcripts; *COX1* and *COX2* TE is decreased more than two-fold in the absence of LRPPRC, while *ND6* TE is increased more than two-fold. Thus, our data suggest that LRPPRC–SLIRP has a role in controlling TE in a transcript-specific manner (Fig. 3b).

To support the role of LRPPRC in mRNA binding, we determined the average length of the mitoribosome-protected fragments using mitoRPF (Fig. 4). In the *LRPPRC*-KO cells, we observed a decrease in the average protected fragment length compared to the WT (Fig. 4). This observation is consistent with the structural data showing the association of LRPPRC with mRNA and the mitoribosome. Previous studies also showed that LRPPRC–SLIRP relaxes the structures of mRNAs[20], potentially exposing them to initiate translation[19]. The average protected fragment length in LSFC cells was smaller than in WT cells, similar to the *LRPPRC*-KO cells (Fig. 4), suggesting that, whereas the mutant protein participates in translation, it does so differently than the WT protein.

### The mitoribosome in complex with LRPPRC–SLIRP is specific to Metazoa

To place the structural data into an evolutionary context, we performed comparative phylogenetic analysis of the proteins involved in the mRNA handoff process. Because the mitochondrial ribosomal RNA (rRNA) is generally reduced in Metazoa[51], we examined whether this loss might coincide with the origin of LRPPRC and its interactors. The orthology database eggNOG[52] and previous analysis[53] indicated that LRPPRC and mS31 are present only in Bilateria, while mS39 occurs only in Metazoa. We then confirmed the results with more sensitive homology detection[54] followed by manual sequence analysis examining domain composition, which put the origin of LRPPRC and mS31 at the root of the Metazoa. Thus, the appearance of these proteins coincides with the loss of parts of the rRNA (Fig. 5a). SLIRP appears to originate slightly later than LRPPRC but its small size makes determining its phylogenetic origin less conclusive.

The correlation between rRNA reduction and protein acquisition is important because the rRNA regions h16 (410–432) and h33–h37 (997–1,118) that bridge the mRNA to the channel entrance in bacteria[11] are either absent or reduced in the metazoan mitoribosome. However, a superposition of the mitoribosome in complex with LRPPRC–SLIRP–mRNA with the *E. coli* expressome[11] shows not only that the nascent mRNA follows a comparable path in both systems but also that the mRNA-delivering complexes bind in a similar location with respect to their ribosomes (Fig. 5b). To test whether protein–protein interactions can explain the conservation, we compared the interface to the *E. coli* expressome[11]. Indeed, in the expressome, NusG binds to uS10 and restrains RNA polymerase (RNAP) motions[11] and, in our structure, uS10m has a related interface between its α2 helix and mS31–mS39, which induces association of these two proteins (Fig. 5c and Extended Data Fig. 8). Yet, most of the interactions rely on a mitochondrion-specific N-terminal extension of uS10m, where it shares a sheet with mS39 through the strand β1, and helices α1, α16 and α18 are further involved in the binding (Extended Data Fig. 8). A similar conclusion can be reached from a comparison to the *Mycoplasma pneumoniae* expressome[13]. Together, this analysis suggests that a specific protein-based mechanism must have evolved in the evolution of the metazoan mitoribosome for mRNA recognition and protection.

### Discussion

LRPPRC is an mRNA chaperone that regulates human mitochondrial transcription and translation and is involved in a neurodegenerative disorder. In this study, we report the cryo-EM structure of LRPPRC–SLIRP in complex with the mitoribosome and characterize its function with respect to mRNA delivery. We identified that LRPPRC, in complex with SLIRP, binds to mRNAs to hand off transcripts to the mitoribosome for translation. The docking of LRPPRC is realized through the mitoribosomal proteins mS39 and the N terminus of mS31 that together recognize eight of the LRPPRC helical repeats. A structural comparison to the unbound state uncovered that the N terminus of mS31 adopts a stable conformation upon LRPPRC association.

Our structure also shows that SLIRP is directly involved in interactions with mRNA. These interactions are supported by a comparison to other RNA-binding proteins that contain RNP domains, similar to SLIRP. SLIRP further stabilizes the architecture of LRPPRC and both are required for mRNA binding. The mRNA is then channeled through a corridor formed with mS39 toward the decoding center.

Although *LRPPRC* KO results in an overall decrease in the steady-state levels of the four OXPHOS complexes that contain mtDNA-encoded subunits, by implementing an RNAseq approach and metabolic labeling assays, we showed that, beyond its role in mRNA stabilization, LRPPRC has differential effects on the translational efficiency of mitochondrial transcripts. Specifically, the syntheses of ND1, ND2, ATP6, COX1, COX2 and COX3 are particularly affected. Thus, the LRPPRC–SLIRP-dependent translation is not the sole regulatory pathway and other mechanisms involving mRNA binding are likely to coexist.

Because mS39 and mS31 are specific to Metazoa, in addition to LRPPRC–SLIRP, the proposed mechanism in which some of the mitochondrial mRNAs are recruited for translation has developed in a coevolutionary manner in Metazoa. However, the presence of large RNA-binding moieties was also reported in association with mitoribosomes in other species[55–59]. Therefore, the principle of regulation by facilitation of molecular coupling might be a general feature, with unique molecular connectors involved in different species.

Overall, these findings define LRPPRC–SLIRP as a regulator of mitochondrial gene expression and explain how its components modulate the function of translation by mRNA binding. Given the challenge of studying mitochondrial translation because of the lack of an in vitro system, the native structures are crucial for explaining fundamental mechanisms. Identification of the components involved enhances our understanding of mitochondrial translation. Together, these studies provide the structural basis for translation regulation and activation in mitochondria.

### Online content

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

## Methods

### Experimental model and culturing

HEK293S-derived cells (T501, originally purchased from Thermo Fisher Scientific) were grown in Freestyle 293 expression medium containing 5% tetracycline-free FBS in vented shaking flasks at 37 °C, 5% $CO_2$ and 120 rpm (550$g$). The cell line tested negative for *Mycoplasma* contamination. The culture was scaled up sequentially, by inoculating at $1.5 \times 10^6$ cells per ml and subsequently splitting at a cell density of $3.0 \times 10^6$ cells per ml. Finally, a final volume of 2 l of cell culture at a cell density of $4.5 \times 10^6$ cells per ml was used for mitochondrion isolation, as described below[61].

### Mitoribosome purification

HEK293S-derived cells were collected from the 2-l culture when the cell density was $4.2 \times 10^6$ cells per ml by centrifugation at 1,000$g$ for 7 min at 4 °C. The pellet was washed and resuspended in 200 ml PBS. The washed cells were pelleted at 1,000$g$ for 10 min at 4 °C. The resulting pellet was resuspended in 120 ml MIB buffer (50 mM HEPES–KOH, pH 7.5, 10 mM KCl, 1.5 mM $MgCl_2$, 1 mM EDTA, 1 mM EGTA, 1 mM dithiothreitol (DTT) and complete EDTA-free protease inhibitor cocktail (Roche)) and allowed to swell in the buffer for 15 min in the cold room by gentle stirring. About 45 ml of SM4 buffer (840 mM mannitol, 280 mM sucrose, 50 mM HEPES–KOH, pH 7.5, 10 mM KCl, 1.5 mM $MgCl_2$, 1 mM EDTA, 1 mM EGTA, 1 mM DTT and 1× complete EDTA-free protease inhibitor cocktail (Roche)) was added to the cells while stirring before pouring into a nitrogen cavitation device kept on ice. The cells were subjected to a pressure of 500 psi for 20 min before releasing the nitrogen from the chamber and collecting the lysate. The lysate was clarified by centrifugation at 800$g$ and 4 °C for 15 min to separate the cell debris and nuclei. The supernatant was passed through a cheesecloth into a beaker kept on ice. The pellet was resuspended in half the previous volume of MIBSM buffer (three volumes MIB buffer + one volume SM4 buffer) and homogenized with a Teflon–glass Dounce homogenizer. After clarification as described before, the resulting lysate was pooled with the previous batch of the lysate and subjected to centrifugation at 1,000$g$ for 15 min at 4 °C to ensure complete removal of cell debris. The clarified and filtered supernatant was centrifuged at 10,000$g$ for 15 min at 4 °C to pellet crude mitochondria. Crude mitochondria were resuspended in 10 ml MIBSM buffer and treated with 200 U of RNase-free DNase (Sigma-Aldrich) for 20 min in the cold room to remove contaminating genomic DNA. Crude mitochondria were again recovered by centrifugation at 10,000$g$ for 15 min at 4 °C and gently resuspended in 2 ml SEM buffer (250 mM sucrose, 20 mM HEPES–KOH, pH 7.5 and 1 mM EDTA). Resuspended mitochondria were subjected to a sucrose density step gradient (1.5 ml of 60% sucrose, 4 ml of 32% sucrose, 1.5 ml of 23% sucrose and 1.5 ml of 15% sucrose in 20 mM HEPES–KOH, pH 7.5 and 1 mM EDTA) centrifugation in a Beckmann Coulter SW40 rotor at 28,000 rpm (139,000$g$) for 60 min. Mitochondria seen as a brown band at the interface of the 32% and 60% sucrose layers were collected and snap-frozen using liquid nitrogen and transferred to −80 °C.

Frozen mitochondria were transferred on ice and allowed to thaw slowly. Lysis buffer (25 mM HEPES–KOH, pH 7.5, 50 mM KCl, 10 mM magnesium acetate, 2% polyethylene glycol octylphenyl ether and 2 mM DTT, 1 mg $ml^{-1}$ EDTA-free protease inhibitors (Sigma-Aldrich)) was added to mitochondria and the tube was inverted several times to ensure mixing. A small Teflon–glass Dounce homogenizer was used to homogenize mitochondria for efficient lysis. After incubation on ice for 5–10 min, the lysate was clarified by centrifugation at 30,000$g$ for 20 min at 4 °C. The clarified lysate was carefully collected. Centrifugation was repeated to ensure complete clarification. A volume of 1 ml of the mitochondrial lysate was applied on top of 0.4 ml 1 M sucrose ($v/v$ ratio of 2.5:1) in thick-walled TLS55 tubes. Centrifugation was carried out at 231,500$g$ for 45 min in a TLA120.2 rotor at 4 °C. The pellets thus obtained were washed and sequentially resuspended in a total volume of 100 μl resuspension buffer (20 mM HEPES–KOH, pH 7.5, 50 mM

KCl, 10 mM magnesium acetate, 1% Triton X-100 and 2 mM DTT). The sample was clarified twice by centrifugation at 18,000$g$ for 10 min at 4 °C. The sample was applied onto a linear 15–30% sucrose gradient (20 mM HEPES–KOH, pH 7.5, 50 mM KCl, 10 mM magnesium acetate, 0.05% $n$-dodecyl-β-D-maltopyranoside and 2 mM DTT) and centrifuged in a TLS55 rotor at 213,600$g$ for 120 min at 4 °C. The gradient was fractionated into 50-μl volume aliquots. The absorption for each aliquot at 260 nm was measured and fractions corresponding to the monosome peak were collected. The pooled fractions were subjected to buffer exchange with the resuspension buffer.

### Cryo-EM data acquisition

A volume of 3 μl of ~120 nM mitoribosome was applied onto a glow-discharged (20 mA for 30 s) holey carbon grid (Quantifoil R2/2, copper, mesh 300) coated with continuous carbon (of ~3-nm thickness) and incubated for 30 s in a controlled environment of 100% humidity and 4 °C. The grids were blotted for 3 s, followed by plunge-freezing in liquid ethane, using a Vitrobot MKIV (Thermo Fisher). The data were collected on FEI Titan Krios (Thermo Fisher) transmission electron microscope operated at 300 keV, using a C2 aperture of 70 μm and a slit width of 20 eV on a GIF quantum energy filter (Gatan). A K2 Summit detector (Gatan) was used at a pixel size of 0.83 Å (magnification of ×165,000) with a dose of 29–32 $e^-$ per $Å^2$ fractionated over 20 frames.

### Cryo-EM data processing

The beam-induced motion correction and per-frame $B$ factor weighting were performed using RELION-3.0.2 (refs. [62,63]). Motion-corrected micrographs were used for contrast transfer function (CTF) estimation with gctf[64]. Unusable micrographs were removed by manual inspection of the micrographs and their respective calculated CTF parameters. Particles were picked in RELION-3.0.2, using reference-free followed by reference-aided particle picking procedures. Reference-free two-dimensional (2D) classification was carried out to sort useful particles from falsely picked objects, which were then subjected to three-dimensional (3D) classification. The 3D classes corresponding to unaligned particles and LSU were discarded and monosome particles were pooled and used for 3D autorefinement yielding a map with an overall resolution of 2.9–3.4 Å for the five datasets. Resolution was estimated using a Fourier shell correlation (FSC) cutoff of 0.143 between the two reconstructed half maps. Finally, the selected particles were subjected to per-particle defocus estimation, beam-tilt correction and per-particle astigmatism correction followed by Bayesian polishing. Bayesian polished particles were subjected to a second round of per-particle defocus correction. A total of 994,919 particles were pooled and separated into 86 optics groups in RELION-3.1 (ref. [65]) on the basis of acquisition areas and date of data collection. Beam tilt, magnification anisotropy and higher-order (trefoil and fourth-order) aberrations were corrected in RELION-3.1 (ref. [65]). Particles with bound P-site tRNA and mRNA that showed comparatively higher occupancy for the unmodeled density potentially corresponding to the LRP-PRC–SLIRP module were pooled and re-extracted in a larger box size of 640 Å. The re-extracted particles were subjected to 3D autorefinement in RELION-3.1 (ref. [65]). This was followed by sequential signal subtraction to remove the signal from the LSU, all of the SSU except the region around mS39 and the unmodeled density, in that order. The subtracted data were subjected to masked 3D classification ($T = 200$) to enrich for particles carrying the unmodeled density. Using a binary mask covering mS39 and all the unmodeled density, we performed local-masked refinement on the resulting 41,812 particles within an extracted sub-volume of 240-Å box size leading to a 3.37-Å resolution map.

### Model building and refinement

At the mRNA channel entrance, a more accurate and complete model of mS39 could be built with 29 residues added to the structure. Improved local resolution enabled unambiguous assignment of residues to the

density, which allowed us to address errors in the previous model. A total of 28 α-helices could be modeled in their correct register and orientation. Furthermore, a 28-residue-long N-terminal loop of mS31 (residues 247–275) along mS39 and a mitochondrion-specific N-terminal extension of uS9m (residues 53–70) approaching mRNA were modeled by fitting the loops into the density maps.

For building the LRPPRC–SLIRP module, the initial model of the full-length LRPPRC was obtained from the AlphaFold2 Protein Structure Database (UniProt P42704). On the basis of the analysis, three stable domains were identified that are connected by flexible linkers (673–983 and 1,035–1,390). We then systematically assessed the domains against the map and the N-terminal region (77–660) could be fitted into the density. The initial model was real-space refined into the 3.37-Å resolution map of the mS39–LRPPRC–SLIRP region obtained after partial signal subtraction using reference restraints in Coot (v.0.9)[66]. The N-terminal region covering residues 64–76 was identified in the density map and allowed us to model 34 helices of LRPPRC (residues 64–644). Helices α1–α29 could be confidently modeled. An additional five helices, as predicted by AlphaFold2 (ref. 35), could be accommodated into the remaining density. After modeling LRPPRC into the map, there was an unaccounted density that fit SLIRP. The initial model of SLIRP was obtained from the AlphaFold2 Protein Structure Database (UniProt Q9GZT3). The unmodeled density agreed with the secondary structure of SLIRP. The model was real-space refined into the density using reference restraints as for LRPPRC in Coot (v.0.9)[66]. Five additional RNA residues could be added to the 3′ terminal of mRNA to account for the tubular density extending from it along the mRNA-binding platform. The A/A P/P E/E state model was rigid-body fitted into the corresponding 2.85-Å resolution consensus map. The modeled LRPPRC was merged with the rigid-body fitted monosome model to obtain a single model of the mitoribosome bound to LRPPRC and SLIRP. The model was then refined against the composite map using PHENIX (v.1.18)[67] (Table 1).

## Phylogenetic analysis

The phylogenetic distribution of proteins was determined by examining phylogeny databases[60], followed by sensitive homology detection to detect homologs outside of the Bilateria. Orthologs were required to have identical domain compositions and Dollo parsimony was used to infer the evolutionary origin of a protein from its phylogenetic distribution. When multiple homologs of a protein were detected in a species, a neighbor-joining phylogeny was constructed to assess monophyly of putative orthologs to the human protein. The short length of the SLIRP candidate protein from *Trichoplax adhaerens* (B3SAC0_TRIAD), which is part of the large RRM family, precludes obtaining a reliable phylogeny to confidently assess its orthology to human SLIRP; therefore, the assessment is tentative.

## TLSMD analysis

The TLSMD analysis[36,37] was performed with the full-length LRPPRC model obtained from the AlphaFold Database (AF-P42704-F1) and the mitochondrion-targeting sequence (residues 1–59) was removed. The model was divided into TLS segments (*N*) and single-chain TLSMD was performed on all atoms using the isotropic analysis model. Instead of using atomic *B* factors, the values for a per-residue confidence score of AlphaFold called the predicted local distance difference test (pLDDT) were used as reference to calculate the least-squared residuals against the corresponding values calculated by TLSMD analysis. This is based on the assumption that local mobility of the model should be inversely correlated with the pLDDT score. AlphaFold pLDDT values and the corresponding calculated values were plotted for every iteration to monitor the improvement in prediction across the length of LRPPRC. The data in Extended Data Fig. 2 are presented for *N* = 4, where segments 1 and 2 (residues 60–373 and 374–649) correspond to the modeled region, whereas segments 3 and 4 correspond to the remaining domains that could not be modeled.

## Helicase sequence analysis

To address the possibility that LRPPRC may serve as a helicase, we inspected the sequence of full-length LRPPRC (UniProt ID P42704). First, we checked the sequence for matches with consensus motifs characteristic of helicases using regular-expression search. The following motifs were searched, GFxxPxxIQ, AxxGxGKT, PTRELA, TPGR, DExD, SAT, FVxT and RgxD (DDX helicases); GxxGxGKT, TQPRRV, TDGML, DExH, SAT, FLTG, TNIAET and QrxGRAGR (DHX helicases); AHTSAGKT, TSPIKALSNQ and MTTEIL (others). Next, we carried out multiple sequence analysis against representative member helicases of the DHX and DDX families to verify the results of the regular-expression sequence search and to find potentially valid weaker matches.

## Human cell lines and cell culture conditions

Human HEK293T embryonic kidney cells (CRL-3216, RRID: CVCL-0063) were obtained from the American Type Culture Collection. The HEK293T *LRPPRC*-KO cell line was engineered in-house and previously reported[31]. The *LRPPRC*-KO cell line was reconstituted with either the WT *LRPPRC* gene[31] or a variant causing LSFC. The LSFC variant carries a single-base change (nucleotide 1119C>T transition), predicting a missense A354V change at a conserved protein residue[47].

Cells were cultured in high-glucose DMEM (Thermo Fisher Scientific, cat. no. 11965092), supplemented with 10% FBS (Thermo Fisher Scientific, cat. no. A3160402), 100 μg ml$^{-1}$ uridine (Sigma, cat. no. U3750), 3 mM sodium formate (Sigma cat. no. 247596) and 1 mM sodium pyruvate (Thermo Fisher Scientific, cat. no. 11360070) at 37 °C under 5% $CO_2$. Cell lines were routinely tested for *Mycoplasma* contamination.

To generate an *LRPPRC*-KO cell line reconstituted with the LSFC variant of the gene, a Myc-DDK-tagged *LRPPRC* open reading frame (ORF) plasmid was obtained from OriGene (cat. no. RC216747). This ORF was then subcloned into a hygromycin resistance-containing pCMV6 entry vector (OriGene, cat. no. PS100024) and used to generate an *LRPPRC*-KO cell line reconstituted with a WT *LRPPRC* gene as reported[31]. To generate the *LRPPRC* LSFC variant carrying the 1119C>T mutation, we used the Q5 site-directed mutagenesis kit from New England Biolabs. Approximately 10 pg of template pCMV6-A-Myc-DDK-Hygro-*LRPPRC* vector was used, along with the primers LSFC-Q5-F 5′-GGAAGATGTAGTGTTGCAGATTTTAC and LSFC-Q5-R 5′-AATTTTTCAGTGACTAAAAGTAAAATG, designed to include the codon to be mutated. After exponential amplification and treatment with kinase and ligase, 2.5 μl of the reaction was transformed into competent *E. coli* cells. Several transformants were selected and their plasmid DNA was purified before sequencing to select the correct pCMV6-A-Myc-DDK-Hygro-*LRPPRC-LSFC* construct.

For transfection of the construct into *LRPPRC*-KO cells, we used 5 μl EndoFectin mixed with 2 μg vector DNA in OptiMEM-I medium according to the manufacturer's instructions. The medium was supplemented with 200 μg ml$^{-1}$ hygromycin after 48 h and drug selection was maintained for at least 1 month.

## Whole-cell extracts and mitochondria isolation

For SDS–PAGE, pelleted cells were solubilized in radioimmunoprecipitation assay (RIPA) buffer (25 mM Tris-HCl, pH 7.6, 150 mM NaCl, 1% NP-40, 1% sodium deoxycholate and 0.1% SDS) with 1 mM PMSF and mammalian protease inhibitor cocktail (Sigma). Whole-cell extracts were cleared by centrifugation at 20,000*g* for 5 min at 4 °C.

Mitochondrion-enriched fractions were isolated from at least ten 80% confluent 15-cm plates as described previously[68–70]. Briefly, the cells were resuspended in ice-cold TKMg buffer (10 mM Tris-HCl, 10 mM KCl and 0.15 mM $MgCl_2$; pH 7.0) and disrupted with ten strokes in a homogenizer (Kimble/Kontes). Using a 1 M sucrose solution, the homogenate was brought to a final concentration of 0.25 M sucrose. A postnuclear supernatant was obtained by centrifugation of the samples twice for 5 min at 1,000*g*. Mitochondria were pelleted by centrifugation

for 10 min at 10,000g and resuspended in a solution of 0.25 M sucrose, 20 mM Tris-HCl, 40 mM KCl and 10 mM $MgCl_2$ (pH 7.4).

## Denaturing and native electrophoresis, followed by immunoblotting

Protein concentration was measured by the Lowry method[71]. First, 40–80 μg of mitochondrial protein extract was separated by denaturing SDS–PAGE in the Laemmli buffer system[72]. Then, proteins were transferred to nitrocellulose membranes and probed with specific primary antibodies to the following proteins: β-actin (dilution 1:2,000; Proteintech, 60008-1-Ig), ATP5A (1:1,000; Abcam, ab14748), CORE2 (1:1,000; Abcam, ab14745), COX1 (dilution 1:2,000; Abcam, ab14705), LRPPRC (dilution 1:1,000; Proteintech, 21175-1-AP), NADH:ubiquinone oxidoreductase subunit A9 (1:1,000; Proteintech, 20312-1-AP), succinate dehydrogenase complex flavoprotein subunit A (1:1,000; Proteintech, 14865-1-AP) or SLIRP (1:1,000; Abcam, ab51523). Horseradish peroxidase-conjugated anti-mouse or anti-rabbit IgGs were used as secondary antibodies (dilution 1:10,000; Rockland). β-Actin was used as a loading control. Signals were detected by chemiluminescence incubation and exposure to X-ray film.

Blue-native PAGE analysis of mitochondrial OXPHOS complexes in native conditions was performed as described previously[73,74]. To extract mitochondrial proteins in native conditions, we pelleted and solubilized 400 μg of mitochondria in 100 μl buffer containing 1.5 M aminocaproic acid and 50 mM Bis-Tris (pH 7.0) with 1% n-dodecyl-β-D-maltoside. Solubilized samples were incubated on ice for 10 min in ice and pelleted at 20,000g for 30 min at 4 °C. The supernatant was supplemented with 10 μl of 10× sample buffer (750 mM aminocaproic acid, 50 mM Bis-Tris, 0.5 mM EDTA and 5% Serva Blue G-250). Native PAGE Novex 3–12% Bis-Tris protein gels (Thermo Fisher) were loaded with 40 μg of mitochondrial proteins. After electrophoresis, the gel was stained with 0.25% Coomassie brilliant blue R250 or proteins were transferred to PVDF membranes using an eBlot L1 protein transfer system (GenScript) and used for immunoblotting.

## Pulse labeling of mitochondrial translation products

To determine mitochondrial protein synthesis, six-well plates were precoated at 5 μg cm$^{-2}$ with 50 μg ml$^{-1}$ collagen in 20 mM acetic acid and seeded with WT or LRPPRC cell lines (two wells per sample per timepoint). Then, 70% confluent cell cultures were incubated for 30 min in DMEM without methionine and then supplemented with 100 μl ml$^{-1}$ emetine for 10 min to inhibit cytoplasmic protein synthesis as previously described[68]. Next, 100 μCi of [$^{35}$S]methionine was added and allowed to incorporate into newly synthesized mitochondrial proteins for increasing times from 15–60-min pulses. Subsequently, whole-cell extracts were prepared by solubilization in RIPA buffer and equal amounts of total cellular protein were loaded into each lane and separated by SDS–PAGE on a 17.5% polyacrylamide gel. Gels were transferred to a nitrocellulose membrane and exposed to a Kodak X-OMAT X-ray film. The membranes were then probed with a primary antibody against β-actin as a loading control. Optical densities of the immunoreactive bands were measured using the Histogram function of the Adobe Photoshop software in digitalized images.

## Whole-cell transcriptomics

Cells were grown to 80% confluency in a 10-cm plate (two plates per sample) and were collected by trypsinization and washed once with PBS before resuspending in 1 ml of Trizol (Thermo Fisher Scientific). RNA was extracted following the Trizol manufacturer's specifications. The aqueous phase was transferred to a new tube and an equal volume of 100% isopropanol and 3 μl of glycogen were added to precipitate the RNA. The sample was incubated at −80 °C overnight and centrifuged at 15,000g for 45 min at 4 °C. RNA was resuspended in 50 μl of RNAse-free water and quantified by measuring absorbance at a wavelength of 260 nm. Then, 2 μg of RNA was sent to Novogene for further processing.

Novogen services included library preparation, RNAseq on an Illumina HiSeq platform according to the Illumina Tru-Seq protocol and bioinformatics analysis. The raw data were cleaned to remove low-quality reads and adaptors using Novogen in-house Perl scripts in Cutadapt[75]. The reads were mapped to the reference genome using the HISAT2 software[76]. The transcripts were assembled and merged to obtain an mRNA expression profile with the StringTie algorithm[77]. The RNAseq data were then normalized to account for the total reads sequenced for each sample (the read depth) and differentially expressed mRNAs were identified by using the Ballgown suite[78] and the DESeq2 R package[79]. GraphPad Prism v.9.0 software was used to prepare the volcano plots.

## MitoRPF

MitoRPF, matched RNAseq and data analysis were performed as previously described[31]. Briefly, human and mouse cell lysates were prepared and mixed in a 95:5 ratio of human to mouse. For mitoRPF, the combined lysates were subjected to RNaseI treatment and fractionated across a linear sucrose gradient. Sequencing libraries were prepared from the monosome fraction after phenol–chloroform extraction. For RNAseq, RNA was extracted from the undigested combined lysate and fragmented by alkaline hydrolysis and sequencing libraries were prepared. Reads were cleaned of adaptors and filtered of rRNA fragments, and PCR duplicates were removed. Read counts were summed across features (coding sequences) using Rsubread feature Counts[80] and then normalized by feature length and mouse spike-in read counts. TE was calculated by dividing spike-in normalized mitoRPF reads per kilobase by spike-in normalized RNAseq reads per kilobase. Values are expressed as the log$_2$ fold change in the LRPPRC-KO cells compared to the LRPPRC rescue cells. MitoRPF and RNAseq data for LRPPRC-KO and LRPPRC reconstituted cell lines were deposited to the Gene Expression Omnibus (GEO) under accession number GSE173283. MitoRPF and RNAseq data for the LSFC reconstituted cell line are deposited to the GEO under accession number GSE221586.

The mitoRPF length distribution was determined from mitochondrial mRNA aligned reads. First, soft-clipped bases were removed using jvarkit[81] and then the frequency for each length was output using SAMtools stats[82].

## Reporting summary

Further information on research design is available in the Nature Portfolio Reporting Summary linked to this article.

## Data availability

The atomic coordinates were deposited to the Research Collaboratory for Structural Bioinformatics PDB and EM maps were deposited in the EM Data Bank under accession numbers 8ANY and EMD-15544. The atomic coordinates used in this study were as follows: PDB 6ZTJ (E. coli 70S–RNAP expressome complex in NusG), PDB 6ZTN (E. coli 70S–RNAP expressome complex in NusG), PDB 1RKJ (human nucleolin), PDB 5WWE (human hnRNPA2/B1) and PDB 1CVJ (PABP). For building the LRPPRC–SLIRP module, the initial model of the full-length LRPPRC was obtained from the AlphaFold2 Protein Structure Database (UniProt P42704). The initial model of SLIRP was obtained from the AlphaFold2 Protein Structure Database (UniProt Q9GZT3). Source data are provided with this paper.

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

## Acknowledgements

We thank S. Aibara and J. Andrell for help with data collection. The research was funded by the Swedish Foundation for Strategic Research (FFL15:0325), Ragnar Söderberg Foundation (M44/16), European Research Council (ERC-2018-StG-805230), Knut and Alice Wallenberg Foundation (2018.0080), National Institutes of Health (NIH; R01-GM123002 to L.S.C. and R35-GM118141 to A.B.). V.S. was supported by the Horizon 2020 Marie Skłodowska-Curie Innovative Training Network (721757), Y.I. was supported by H2020-MSCA-IF-2017 (799399-Itohribo) and C.M. was supported by the Eunice Kennedy Shriver National Institute Of Child Health and Human Development of the NIH under award number F30HD107939. The SciLifeLab cryo-EM facility is funded by the Knut and Alice Wallenberg, Family Erling Persson and Kempe foundations. The content is solely the responsibility of the authors and does not necessarily represent the official views of the NIH.

## Author contributions

V.S. collected cryo-EM data, processed the data and built the models. V.S., Y.I. and A.A. performed structural analysis. C.M., F.F. and A.B. performed mitochondrial translation, OXPHOS and RNAseq analysis. I.S., M.C. and L.S.C. performed mitoRPF and RNAseq analysis. V.S., M.H. and A.A. performed evolutionary analysis. A.A. wrote the manuscript. All authors contributed to data interpretation and manuscript writing.

## Competing interests

The authors declare no competing interests.

## Additional information

**Extended data** is available for this paper at https://doi.org/10.1038/s41594-024-01365-9.

**Correspondence and requests for materials** should be addressed to Yuzuru Itoh, L. Stirling Churchman, Antoni Barrientos or Alexey Amunts.

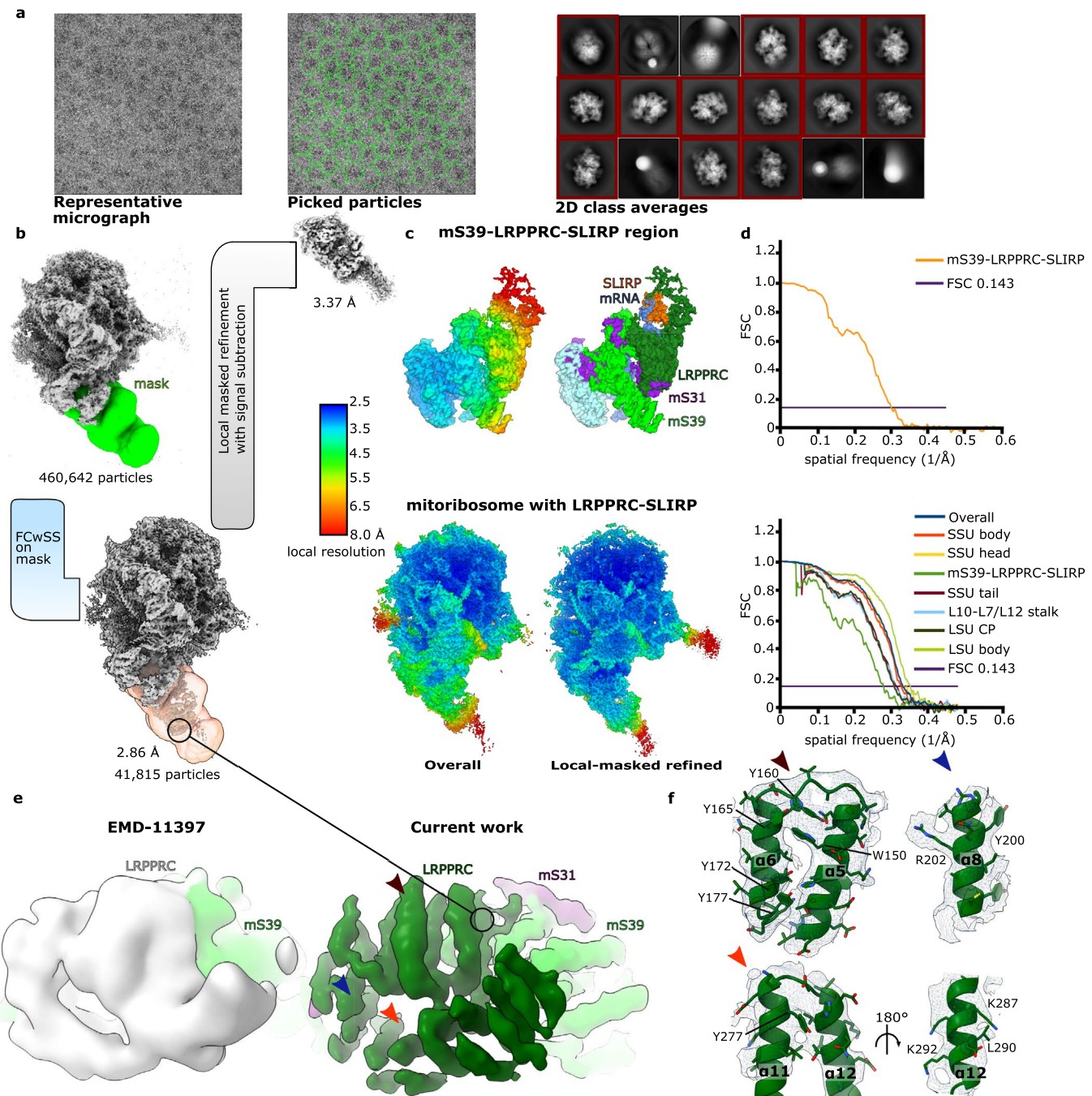

**Extended Data Fig. 1 | Cryo-EM data processing and map quality for mS39-LRPPRC-SLIRP region. a.** A representative micrograph with picked particles (green circles) and 2D class averages with mixed LSU and monosome particles (marked by red boxes). **b.** Focused 3D-classification with signal subtraction using mask around mS39-LRPPRC-SLIRP region (transparent orange) of mitoribosome particles to identify LRPPRC-SLIRP containing monosome particles (2.86 Å overall resolution), followed by masked refinement with signal subtraction on mS39-LRPPRC-SLIRP region to improve the local resolution. **b.** The mS39-LRPPRC-SLIRP map is shown colored by local resolution (top left)

and by proteins assigned to the density (top right). The consensus map (bottom left) and the masked refined maps shown as a single composite map colored by local resolution (bottom right). **c.** Fourier shell correlation curves for the post-subtraction masked refined mS39-LRPPRC-SLIRP map (top) and individual masked refined maps. **e.** Map comparison for LRPPRC region between our work and EMD-11397. The map has been Gaussian filtered for better visibility. **f.** Density shown as mesh around helices α5-6, 8 and 11-12. Corresponding regions are indicated with arrows in panel (e).

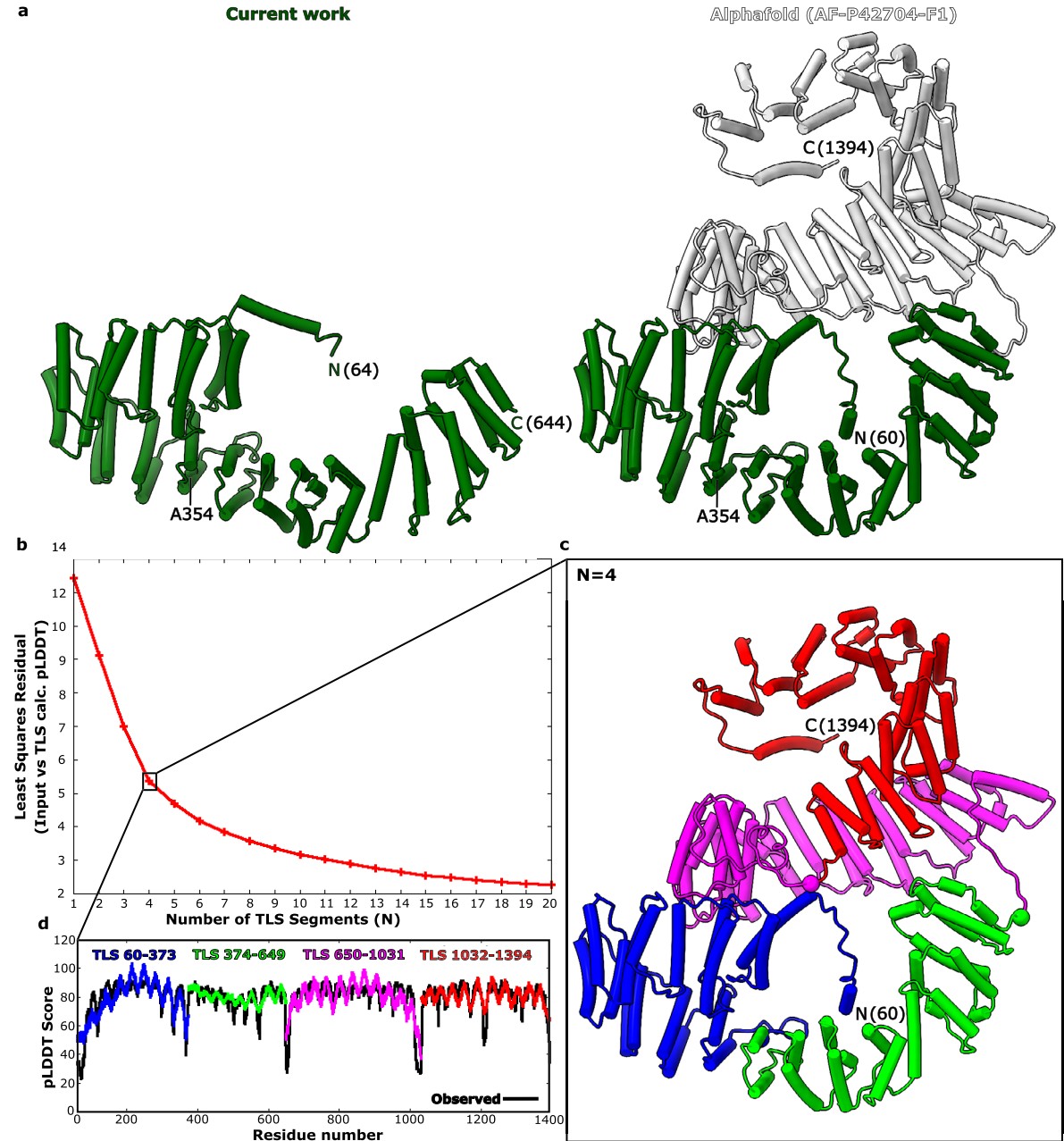

**Extended Data Fig. 2 | AlphaFold model and TLSMD analysis of LRPPRC.**
**a**. The modeled region of LRPPRC (residues 64-644) is compared with the AlphaFold model (AF-P42704-F1) of full length (right). The modeled region is green, the unmodeled is white. The position of LSFC variant (A354V) is indicated. **b**. TLSMD analysis of the AlphaFold model of LRPPRC up to 20 TLS segments

(N). Graph plots least-square residuals assigned per-residue confidence score values (pLDDT) versus those calculated by TLS analysis. **c**. Model colored by TLS segments for N = 4. Regions between the segments with high pLDDT values correspond to loop regions and are shown as spheres **d**. Comparison of AlphaFold assigned versus calculated pLDDT values at N = 4.

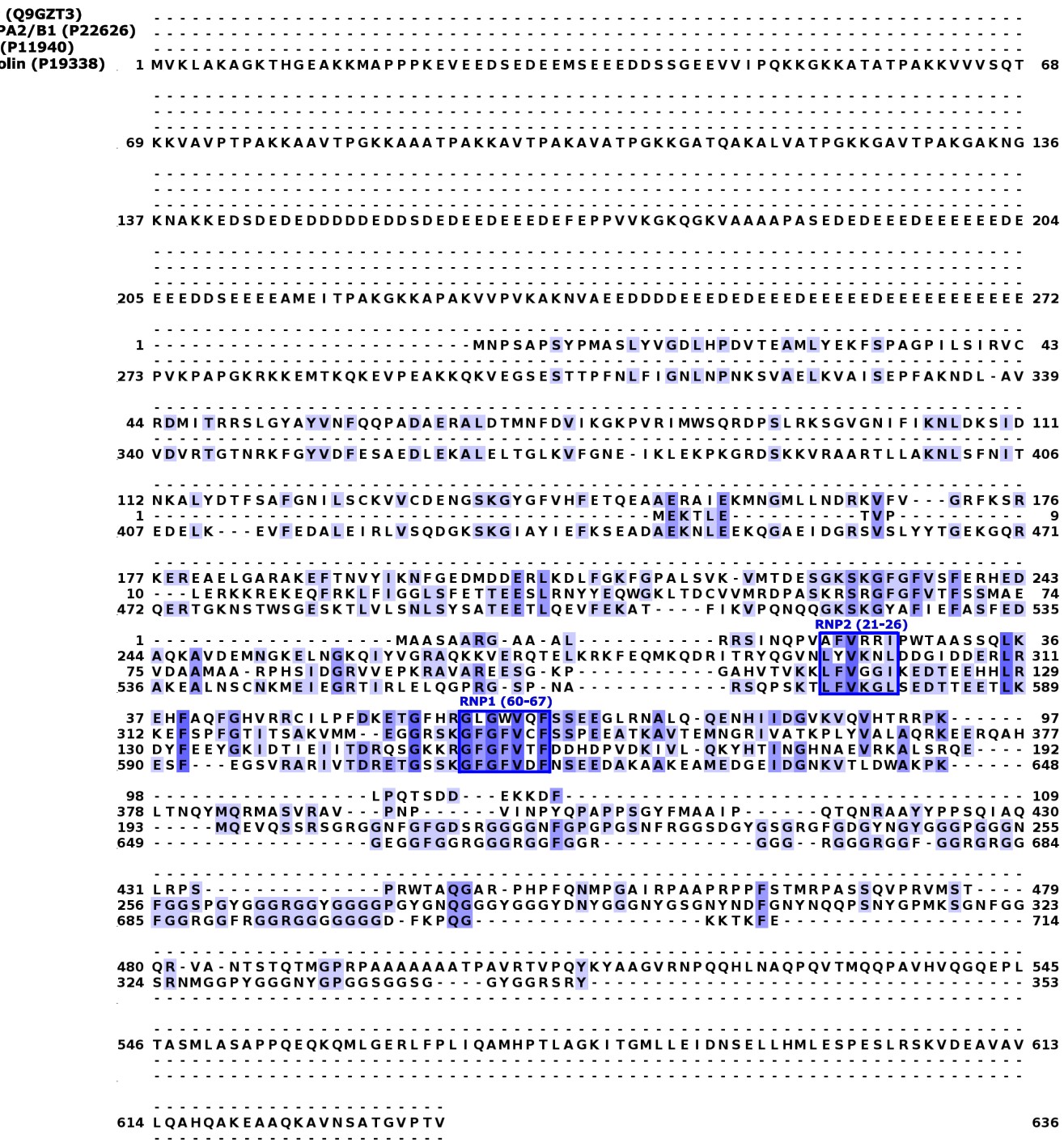

**Extended Data Fig. 3 | Multiple sequence alignment between SLIRP and representative RRM containing proteins.** Alignment of SLIRP with representative RRM family proteins, heterogeneous nuclear ribnucleoproteins (hnRNPA2/B1), poly-A binding protein (PABP), and nucleolin shows conservation of submotifs RNP1 and RNP2 highlighted and indicated by corresponding residue numbers in SLIRP. Individual sequences are marked by residue numbers in the beginning and end and residues are colored by present identity.

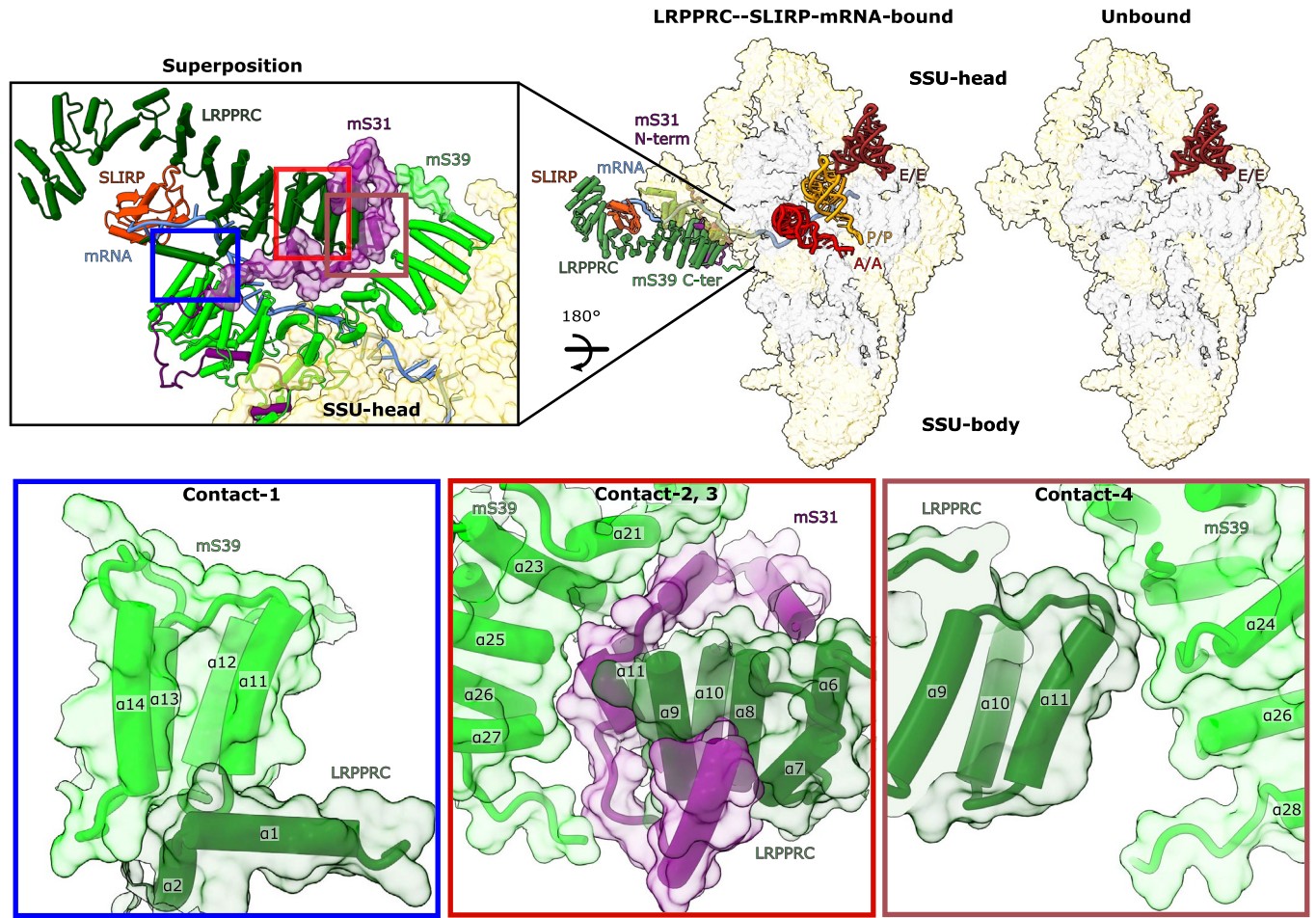

**Extended Data Fig. 4 | LRPPRC-SLIRP contacts with the SSU head.** Comparison of SSU from mitoribosome:LRPPRC-SLIRP complex with SSU from E-site tRNA bound monosome. Zoom-in shows N-terminal region of mS31 and C-terminal loop of mS39 (in surface) stabilized by LRPPRC. Contact regions of LRPPRC with mS31 and mS39 shown in cartoon and surface representations.

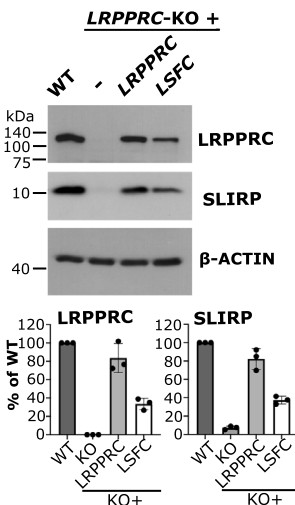

**Extended Data Fig. 5 | Reconstitution of the *LRPPRC*-KO with wild-type and LSCF variants of LRPPRC.** Immunoblot analysis to estimate the steady-state levels of LRPPRC and SLIRP in the indicated cell lines. β-ACTIN was used as a loading control. The images were digitized, and the specific signals were quantified using the histogram function of Adobe Photoshop from three independent repetitions.

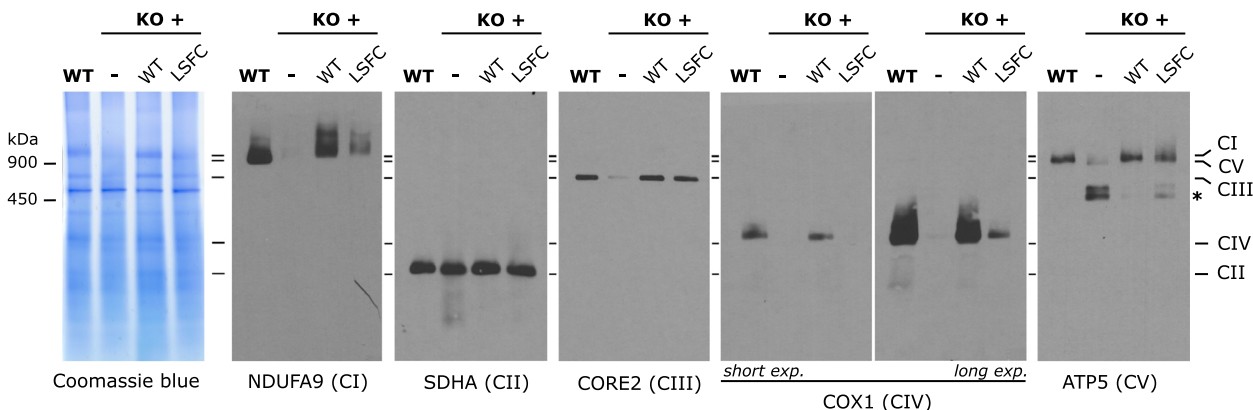

**Extended Data Fig. 6 | Mitochondrial protein synthesis is altered in *LRPPRC*-KO cells.** Blue-native PAGE analyses in WT, *LRPPRC*-KO, and KO + WT cell lines. Intact respiratory complexes were extracted from purified mitochondria using 1% n-dodecyl β-D-maltoside. An asterisk indicates the ATPase (CV) $F_1$ module that accumulates due to the low levels of the mitochondrion-encoded $F_0$ module subunits ATP6 and ATP8.

a

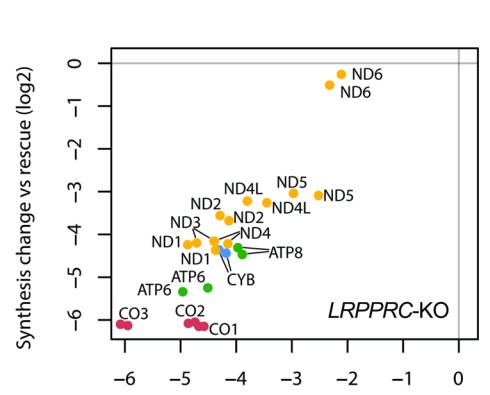

b

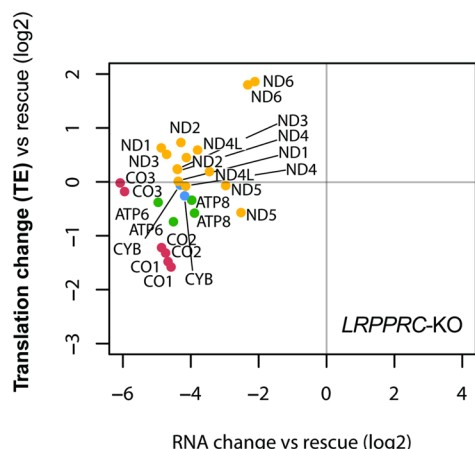

**Extended Data Fig. 7 | Mitochondrial translation efficiency is differentially affected in *LRPPRC*-KO cells. a**, Change in inferred protein synthesis (mitoribosome profiling coverage) versus RNA abundance in *LRPPRC*-KO cells compared to LRPPRC-reconstituted cells ("rescue"). Mitoribosome profiling data and RNA-seq data were normalized using a mouse lysate spike-in control[31].

**b**, Translation efficiency (TE) was calculated from spike-in normalized values (mitoribosome profiling / RNA-seq) and again plotted against change in RNA abundance so that the x-axis values are the same as in (a). Biological replicates are shown as individual points. The mitochondrial transcripts are color-coded as in Fig. 3.

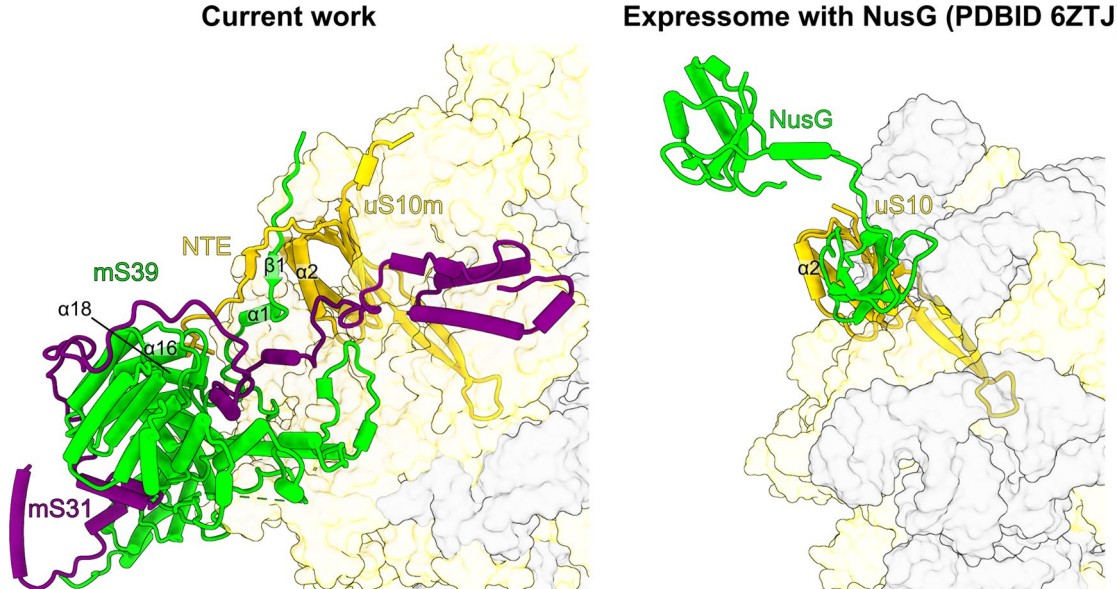

**Extended Data Fig. 8 | Close-up view of uS10m interactions with mS31-mS39.** Interface between uS10m with mS31-mS39 that serve as the platform for LRPPRC-SLIRP is similar to that formed between uS10 and NusG that binds RNA polymerase in bacterial expressome[11].

# Reporting Summary

## Statistics

For all statistical analyses, confirm that the following items are present in the figure legend, table legend, main text, or Methods section.

| n/a | Confirmed | |
|---|---|---|
| ☐ | ☒ | The exact sample size (*n*) for each experimental group/condition, given as a discrete number and unit of measurement |
| ☐ | ☒ | A statement on whether measurements were taken from distinct samples or whether the same sample was measured repeatedly |
| ☒ | ☐ | The statistical test(s) used AND whether they are one- or two-sided<br>*Only common tests should be described solely by name; describe more complex techniques in the Methods section.* |
| ☒ | ☐ | A description of all covariates tested |
| ☒ | ☐ | A description of any assumptions or corrections, such as tests of normality and adjustment for multiple comparisons |
| ☐ | ☒ | A full description of the statistical parameters including central tendency (e.g. means) or other basic estimates (e.g. regression coefficient) AND variation (e.g. standard deviation) or associated estimates of uncertainty (e.g. confidence intervals) |
| ☒ | ☐ | For null hypothesis testing, the test statistic (e.g. *F*, *t*, *r*) with confidence intervals, effect sizes, degrees of freedom and *P* value noted<br>*Give P values as exact values whenever suitable.* |
| ☒ | ☐ | For Bayesian analysis, information on the choice of priors and Markov chain Monte Carlo settings |
| ☒ | ☐ | For hierarchical and complex designs, identification of the appropriate level for tests and full reporting of outcomes |
| ☒ | ☐ | Estimates of effect sizes (e.g. Cohen's *d*, Pearson's *r*), indicating how they were calculated |

*Our web collection on statistics for biologists contains articles on many of the points above.*

## Software and code

Policy information about availability of computer code

| Data collection | The datasets were collected EPU 1.9 software on FEI Titan Krios (FEI/Thermofischer) transmission electron microscope operated at 300 keV with a slit width of 20 eV on a GIF quantum energy filter (Gatan). A K2 Summit detector (Gatan) was used at a pixel size of 0.83 Å (magnification of 165,000x) with a dose of 29-32 electrons/Å2 fractionated over 20 frames.  A defocus range of -0.6 to -2.8 μm was used. |
|---|---|
| Data analysis | Movie frames were aligned and averaged by global and local motion corrections by the program RELION-3.0.2. Contrast transfer function (CTF) parameters were estimated by gctf. Particles were picked, 2D and 3D classified by RELION 3.0.2. Particle separation into optics groups was done in RELION-3.1. Beam-tilt, magnification anisotropy and higher-order (trefoil and fourth-order) aberrations were corrected in RELION-3.1. The models were manually built with Coot 0.9 and stereochemical refinement was performed using phenix.real_space_refine in the PHENIX 1.18 suite. Optical densities of the immunoreactive bands were measured using the Histogram function of the Adobe Photoshop 22.0.0 software in digitalized images. |

For manuscripts utilizing custom algorithms or software that are central to the research but not yet described in published literature, software must be made available to editors and reviewers. We strongly encourage code deposition in a community repository (e.g. GitHub). See the Nature Portfolio guidelines for submitting code & software for further information.

## Data

Policy information about <u>availability of data</u>

All manuscripts must include a <u>data availability statement</u>. This statement should provide the following information, where applicable:

- Accession codes, unique identifiers, or web links for publicly available datasets
- A description of any restrictions on data availability
- For clinical datasets or third party data, please ensure that the statement adheres to our <u>policy</u>

The atomic coordinates were deposited in the RCSB Protein Data Bank, and EM maps have been deposited in the Electron Microscopy Data bank under accession numbers 8ANY and EMD-15544. The atomic coordinates that were used in this study: 6ZTJ (E.coli 70S-RNAP expressome complex in NusG); 6ZTN (E.coli 70S-RNAP expressome complex in NusG); 1RKJ (human Nucleolin); 5WWE (human hnRNPA2/B1); 1CVJ (Poly-adenylate binding protein, PABP). For building LRPPRC-SLIRP module, the initial model of the full length LRPPRC was obtained from AlphaFold2 Protein Structure Database (Uniprot ID P42704). The initial model of SLIRP was obtained from Alphafold2 Protein Structure Database (Uniprot ID Q9GZT3).

## Human research participants

Policy information about <u>studies involving human research participants and Sex and Gender in Research.</u>

| | |
|---|---|
| Reporting on sex and gender | N/A |
| Population characteristics | N/A |
| Recruitment | N/A |
| Ethics oversight | N/A |

Note that full information on the approval of the study protocol must also be provided in the manuscript.

# Field-specific reporting

Please select the one below that is the best fit for your research. If you are not sure, read the appropriate sections before making your selection.

☒ Life sciences ☐ Behavioural & social sciences ☐ Ecological, evolutionary & environmental sciences

For a reference copy of the document with all sections, see nature.com/documents/nr-reporting-summary-flat.pdf

# Life sciences study design

All studies must disclose on these points even when the disclosure is negative.

| | |
|---|---|
| Sample size | A total of 994,919 particles were pooled and analyzed. No statistical analyses has been performed. The number of cryo-EM particles in the single dataset collected was the number of particles available. No predetermined sample size was used for other experiments. |
| Data exclusions | For cryo-EM structure determination, particles that were not mitochondrial ribosome were discarded by classification, since they cannot contribute to reconstruction. |
| Replication | Cryo-EM structures were successfully obtained from three preliminary datasets. |
| Randomization | Cryo-EM map resolution estimates by Fourier Shell Correlation were performed using half-maps from random half-sets. |
| Blinding | Raw micrographs or particle images are not categorical data. Particles are randomly assigned into half-sets for image processing; hence no blinding is applicable. |

# Reporting for specific materials, systems and methods

We require information from authors about some types of materials, experimental systems and methods used in many studies. Here, indicate whether each material, system or method listed is relevant to your study. If you are not sure if a list item applies to your research, read the appropriate section before selecting a response.

## Materials & experimental systems

| n/a | Involved in the study |
|---|---|
| ☐ | ☒ Antibodies |
| ☐ | ☒ Eukaryotic cell lines |
| ☒ | ☐ Palaeontology and archaeology |
| ☒ | ☐ Animals and other organisms |
| ☒ | ☐ Clinical data |
| ☒ | ☐ Dual use research of concern |

## Methods

| n/a | Involved in the study |
|---|---|
| ☒ | ☐ ChIP-seq |
| ☒ | ☐ Flow cytometry |
| ☒ | ☐ MRI-based neuroimaging |

## Antibodies

Antibodies used

Primary antibodies against β-ACTIN (dilution 1:2,000; Proteintech; Rosemont, IL; 60008-1-Ig), ATP5A (1:1000; Abcam; Cambridge, MA; ab14748), CORE2 (1:1,000; Abcam; Cambridge, MA; ab14745), COX1 (dilution 1:2,000; Abcam; Cambridge, MA; ab14705), LRPPRC (dilution 1:1,000; Proteintech; Rosemont, IL; 21175-1-AP), NDUFA9 (1:1000; Proteintech; Rosemont, IL; 20312-1-AP), SDHA (1:1,000; Proteintech; Rosemont, IL; 14865-1-AP) or SLIRP (1:1000; Abcam; Cambridge, MA; ab51523). Horseradish peroxidase-conjugated anti-mouse or anti-rabbit IgGs were used as secondary antibodies (dilution 1:10,000; Rockland; Limerick, PA).

Validation

*Describe the validation of each primary antibody for the species and application, noting any validation statements on the manufacturer's website, relevant citations, antibody profiles in online databases, or data provided in the manuscript.*

## Eukaryotic cell lines

Policy information about cell lines and Sex and Gender in Research

Cell line source(s)

HEK293S-derived cells line T501 was originally purchased from Thermofisher Scientific.

Authentication

No authentication is required as these cells are Zeocin and Blasticidin resistant.

Mycoplasma contamination

Cell line tested negative for mycoplasma contamination.

Commonly misidentified lines
(See ICLAC register)

No commonly misidentified cell lines were used in the study.

