## [Peer Review File · Nature Structural & Molecular Biology]

Peer Review Information

Manuscript Title: Structural basis of LRPPRC-SLIRP-dependent translation by the mitoribosome

Corresponding author name(s): Alexey Amunts, Antoni Barrientos, Stirling Churchman, Yuzuru Itoh

Reviewer Comments & Decisions:

Decision Letter, initial version:

Message: 14th Sep 2022

Dear Dr. Amunts,

Thank you again for submitting your manuscript "Activation mechanism of mitochondrial translation by LRPPRC-SLIRP". We now have comments (below) from the 2 reviewers who evaluated your paper. In light of those reports, we remain interested in your study and would like to see your response to the comments of the referees, in the form of a revised manuscript.

You will see that both reviewers ask for additional clarification about the modeling and interpretation, especially of the dynamic regions. Please be sure to address/respond to all concerns of the referees in full in a point-by-point response and highlight all changes in the revised manuscript text file. If you have comments that are intended for editors only, please include those in a separate cover letter.

We expect to see your revised manuscript within 6 weeks. If you cannot send it within this time, please contact us to discuss an extension; we would still consider your revision, provided that no similar work has been accepted for publication at NSMB or published elsewhere.

Reporting Summary:

When submitting the revised version of your manuscript, please pay close attention to our [href="https://www.nature.com/nature-portfolio/editorial-policies/image-integrity">Digital Image Integrity Guidelines](https://www.nature.com/nature-portfolio/editorial-policies/image-integrity). and to the following points below:

Please note that all key data shown in the main figures as cropped gels or blots should be presented in uncropped form, with molecular weight markers. These data can be aggregated into a single supplementary figure item. While these data can be displayed in a relatively informal style, they must refer back to the relevant figures. These data should be submitted with the final revision, as source data, prior to acceptance, but you may want to start putting it together at this point.

Data availability: this journal strongly supports public availability of data. All data used in accepted papers should be available via a public data repository, or alternatively, as Supplementary Information. If data can only be shared on request, please explain why in your Data Availability Statement, and also in the correspondence with your editor. Please note that for some data types, deposition in a public repository is mandatory - more

information on our data deposition policies and available repositories can be found below:
<https://www.nature.com/nature-research/editorial-policies/reporting-standards#availability-of-data>

Nature Structural & Molecular Biology is committed to improving transparency in authorship. As part of our efforts in this direction, we are now requesting that all authors identified as 'corresponding author' on published papers create and link their Open Researcher and Contributor Identifier (ORCID) with their account on the Manuscript Tracking System (MTS), prior to acceptance. This applies to primary research papers only. ORCID helps the scientific community achieve unambiguous attribution of all scholarly contributions. You can create and link your ORCID from the home page of the MTS by clicking on 'Modify my Springer Nature account'. For more information please visit please visit www.springernature.com/orcid.

[Redacted]

Sincerely,
Sara

Sara Osman, Ph.D.
Associate Editor
Nature Structural & Molecular Biology

Referee expertise:

Referee #1: Cryo-EM, translation

Referee #2: Cryo-EM, transcription-translation coupling

Reviewers' Comments:

Reviewer #1:

Remarks to the Author:

The manuscript by Singh et al. describes the results of a structural analysis of a novel complex linking transcription and translation in human mitochondria, using cryo-electron microscopy. Indeed, the authors reveal the first structure of the human mitochondrial ribosome (mitoribosome) in complex with LRPPRC (leucine-rich pentatricopeptide repeat-containing protein) and SLIRP (SRA stem-loop-interacting RNA-binding protein), which are both proteins implicated in the interaction with mt-mRNA. In their structure, the authors show how the mt-mRNA is delivered after its transcription to the mitoribosome that remains attached to the inner mt-membrane.

Although brief, the manuscript is well written and the figures are crystal-clear. The refined structures appear to be of quality based on their statistics and references are sufficient.

The presented structure will be of great interest to the communities of translation regulation, mitochondrial gene expression and the ribosome more broadly. The thorough cryo-EM analysis is also interesting and similar data-processing workflow can be applied to other complexes. I warmly recommend its publication but would however have few suggestions and minor comments:

The authors assign the additional densities to LRPPRC and SLIRP. However, it is not mentioned anywhere in the manuscript how the authors came to this assignment. For instance, mass spectrometry analysis could shed light on the complex composition. Alternatively, the resolution of the map could also be sufficient for accurate assignment when sidechains are clearly resolved, can the authors show some representative portions that allowed them to interpret their structure? Finally, AlphaFold models and their cross-correlation to the map could also suffice, but perhaps the authors need to include some additional supplementary figures/panels to demonstrate their assignment.

The authors describe the mRNA densities in which some 42 residues were modeled. However, nowhere in the manuscript they show this very crucial detail on their structure! In my opinion, this is a very important point that merits some more demonstration and highlight. For example, the authors could show fitting of a part of this mRNA portion in its densities, also surface representation of the segmented densities at this region, so the reader can appreciate the resolution and the reliability of this part of the structure.

The authors refer to an Extended Data Figure 5c, which doesn't exist (Page 3 line 86).

Extended Data Figure 2c requires some explanation, but perhaps it is not the most efficient way to represent the complex interactions at the SSU beak.

Yaser Hashem

Reviewer #2:

Remarks to the Author:
Overview

Singh et al. report a structural model of the mitochondrial ribosome bound to mRNAs it is translating and the regulatory proteins LRPPRC-SLIRP associated with the mRNA entrance channel. This structure is of great importance for both the insight it offers into fundamental gene expression mechanisms and its relevance to human disease. The use of a native purification scheme from mitochondria provides strong support that the presented architecture is of physiological relevance, and this is a technical achievement to be applauded.

There are no flaws that prohibit publication, and predictions made are fully testable. The work has impact beyond the immediate field of mitochondrial translation: parallels are drawn to molecular mechanisms of bacterial gene expression recently presented that will attract broad interest. The methodology is cutting-edge and the work will be appreciated by many interested in structure determination by cryo-EM. The length of text appears appropriate; as detailed below, an increase in waiting towards interpretation of the results and away from describing the model is needed in this reviewer's opinion.

Major concern

The authors provide all the data and figures that are typically essential for a reader to evaluate the quality of a structural model. However, as noted in the manuscript, the region of this complex of greatest interest (LRPPRC, SLIRP, mS31, mS39 and protruding mRNA) is dynamic. The depth of structural information available for each of these components should be presented more clearly as the conclusions that are drawn from the structure require an indication of their degree of confidence. This concern is deemed 'major' as this clarification would substantially improve the ability of the field to appreciate which questions have been addressed comprehensively here, and what remains to be further characterised.

Below are some specific suggestions this reviewer believes would improve the manuscript in this regard:

- The focus of the manuscript are the structural features adjacent to the mRNA entrance channel, which consists of LRPPRC, SLIRP, mS31 and mS39. The local resolution of this region is reported as 3.4 Å, but there is significant variation in this region as indicated in ExFig1a. To clarify which features are well-resolved and which are not, the authors should consider (1) presenting a map-model overlay in Figure 1 (rather than model alone), (2) labelling the local resolution map in ExFig1a with which proteins correspond to which density. This is also important as it avoids the apparent discrepancy that the representative density shown ExFig1c does not appear to <3.5 Å (to this reviewer's eyes). Although the presentation of local B-factor in Fig1 conveys some of this information, it does not make it clear what regions can be interpreted at level of amino acid contacts or specific nucleotides, and the manuscripts conclusions rely on this.
- The model of mRNA in the structure is presented with apparent confidence: 42 residues modelled and residues 31-32 recognised by specific residues of LRPPRC. Presenting a map-model overlay of the mRNA chain alone would indicate to readers the confidence in this aspect of the model. For some fraction of the mRNA chain it is likely the resolution prevents unambiguous modelling; given the capacity for ssRNA to expand or contract, could the authors instead indicate a range of RNA lengths that may be sequestered by the channel of proteins and describe the model as 'representative'? This conclusion is

particularly important given a primary role of LRPPRC-SLIRP is in protecting/unwinding RNA: the field would benefit from knowing the best estimate of how many nucleotides are protected by the proteins.

The focus of this manuscript, as indicated in the title, is on a translation activation mechanism. However, while clear detail is provided in describing the structural model, very little text is devoted to explaining this mechanism. Could the authors elaborate on the connection between mRNA 'hand-off' and translation activation:

- Is the unfolding of mRNA secondary structure the primary activating role of LRPPRC, as was proposed for the NusG in the coupled E. coli expressome (PMID: 32820062)? The E. coli ribosome has intrinsic RNA helicase activity that has been localised to specific residues (PMID: 15652481) – are these conserved in the human mitoribosome, and if not perhaps LRPPRC-SLIRP provides essential helicase activity?
- It is not clear whether the authors propose the LRPPRC-SLIRP complex is loaded onto the ribosome with the mRNA or if a complex with the ribosome pre-exists arrival of the mRNA. It is indicated to bind co-transcriptionally (line 36) but, in contrast, a suggestion is made that it is part of SSU biogenesis (line 146). If the latter is true, LRPPRC-SLIRP could be viewed as a recruitment factor that increases the affinity of mitoribosomes for mRNA (line 71), and this represents a distinct translation activation mechanism. This could be highlighted.

A conclusion on the mechanistic model should be included in the Abstract.

Minor concerns

The following list contains suggestions that this reviewer believe would significantly aid the clarity of the manuscript:

- Line 19: how does this complex 'act as a carrier of genetic information' in a way that it is different to how mRNA alone does. Perhaps the authors mean the complex provides efficiency in gene expression?
- Lines 29-32: The wording of the opening sentences implies LRPPRC exists in mitochondria because it provides mRNA protection that in bacteria is provided by transcription-translation coupling (TTC). If this is intentional it is not a correct comparison: most translation in bacteria occurs on mRNA that has been released from RNAP and is not within the nucleoid; further, bacteria contain other proteins that confer mRNA stability. Could the authors take a different approach to the introduction?
- Lines 36-37: could this sentence be clarified: LRPPRC both stabilizes and destabilizes mRNA secondary structures? Is it possible the authors mean to remove the word 'structure' in line 37? The term 'global RNA chaperone' used by Siira et al. may also be useful.
- Line 45: missing full stop.
- Line 47: the phrasing 'might' undersells the known cooperation between LRPPRC and SLIRP. The interaction of LRPPRC and SLIRP has been studied in some detail in reference [20] and this could be indicated here.
- Line 54: should state 'entry channel' to distinguish from mRNA exit channel.
- Line 56: local resolution of which specific region should be indicated in these brackets.
- Line 56: 'trapped' should perhaps be changed to 'visualised' or similar: no method of trapping the complex was employed other than the freezing process.
- Line 58: could the authors indicate the exact residue numbers included in the modelled 'N-terminal domain' and unmodelled 'C-terminal domain'.

- Line 61: 'outspreads' could be changed to a more common word: e.g. 'projects' or 'protrudes'.
- Line 62: the use of terms 'NES' vs 'NES-rich' should be consistent with Fig. 1. Can the authors confirm it is necessary to introduce the terms 'head' and 'tail' when it appears these domains have names already?
- Fig.1c: could this include a domain architecture diagram making it clear that a C-terminal domain is present in LRPPRC that was not modelled?
- Line 63: 'completion' rather than completing.
- Line 64: the manuscript would benefit from a justification of why this density can be confidently assigned to SLIRP. Was mass spectrometry analysis performed to confirm SLIRP is present in reasonable abundance? Was the interaction interface previously determined (ref [20]) used to assign the density? Does AlphaFold predict a LRPPRC-SLIRP interaction consistent with this?
- Line 69: from the start of the E-site?
- Line 74: crosslinking mass spectrometry data and mutational analysis
- Line 76: for clarity: 'contributes to a corridor for the mRNA that extends that formed by mS31 and mS39' or similar.
- Line 80: it should be stated here or elsewhere that SLIRP is an RRM-containing protein. Could the authors comment on whether the interface of SLIRP that contacts mRNA parallels other known RRM-RNA interactions? Were specific basic or aromatic residues identified that are predicted to be essential for RNA-binding activity?
- Line 86: no ExFig5 was provided.
- Line 95: should be: hand-off of the mRNA. The term 'handing-off' is used throughout the manuscript, but are 'delivery' or 'transfer' more common terms for this idea? If not, this should be 'hand-off' for a noun.
- Line 98: the residue numbers of the arginines should be stated. A figure for this would be useful.
- Line 100: 'analysis of recombinant LRPPRC' would be more specific.
- Line 106: 'contact-2' has not been introduced before this line (nor contact-1).
- Line 120: metazoa should be capitalized
- Line 126: 'therefore' rather than 'therewith'
- Line 130: the mRNA entrance channel in bacteria is primarily defined by proteins S3, S4 and S5. The mentioned rRNA regions border these. Could the authors comment on whether S3,S4 and S5 are conserved in their structural roles within the mitoribosome, and if they are why is the loss of the rRNA is worth noting here?
- Line 132: The chosen expressome structure for comparison is not the best: the mRNA path was clearly resolved only in the structure with a shorter mRNA (PDB: 6ZTJ). Here, the mRNA travels across the surface of ribosomal protein S3 (see PMID: 32820062 figure 2). Do the authors suggest the path is comparable in the involvement of S3? It appears the similarity to the expressome is more general: a positively charged surface on the exterior of the ribosome immediately adjacent to the mRNA entrance channel. The authors should also cite a similar channel mechanism formed by NusA in the M. pneumoniae expressome (PMID: 32732422 Fig. 3).
- Line 139: 'mito-specific' should be mitoribosome-specific.
- Line 141: a mechanisms of specific mechanisms. This could be described as a convergent evolution of translation mechanisms with reference to bacterial transcription-translation coupling factors.
- Line 143: 'the' central regulator of gene expression sounds like an overstatement: 'a' central regulator?
- Line 146: citations appear to be the wrong order: 27-26.
- Line 148-149: the wording of this requires revision to be clear what is meant.

- Could the positions of mutations in LRPPRC/SLIRP known to be disease-related be mapped on the structural model in an Extended data figure?
- Reference 6 should be 2012: could the authors check accuracy of all references.

Author Rebuttal to Initial comments

We thank the Reviewers for their kind comments and for taking the time to provide constructive suggestions on how to improve the study, its readability and presentation. We addressed all the requests and added experimental data that clarify the role of LRPPRC. Specifically, we performed RNAseq, metabolic labeling and mitoribosome profiling that showed a major influence on ND1, ND2, ATP6, COX1, COX2, and COX3 mRNA translation efficiency. These data suggest that LRPPRC-SLIRP does not preexist on the mitoribosome as its structural element but rather acts in recruitment of specific mRNAs to modulate their translation. In the revised version, we added the following figures:

- *Fig. 2 showing overview of density and comparison between RRM containing proteins.*
- *Fig. 3b,c showing RNAseq and metabolic labeling with [³⁵S]-labeled methionine of newly-synthesized mitochondrial polypeptides.*
- *Fig. 4 showing that mitochondrial translation efficiency is decreased in LRPPRC-KO cells from mitoribosome profiling data and RNA-seq, as well as heat map showing the length distribution for reads mapping to mitochondrial mRNAs.*
- *Extended Data Fig. 2 with AlphaFold model and TLSMD analysis of LRPPRC.*
- *Extended Data Fig. 3 with multiple sequence alignment between SLIRP and representative RRM containing proteins.*
- *Extended Data Fig. 4 showing reconstitution of the LRPPRC-KO with wild-type and LSFC variants of LRPPRC.*
- *Extended Data Fig. 5 showing how mitochondrial protein synthesis is altered in LRPPRC-KO cells.*

In the text, to better organise the newly added data, we now arranged the Results in five sections:

- *Structure determination of LRPPRC-SLIRP bound to the mitoribosome*
- *SLIRP is stably associated with mRNA and LRPPRC on the SSU*
- *LRPPRC is recruited for translation of specific mRNAs*
- *LRPPRC-SLIRP hands-off the mRNA to mS31-mS39, channeling it for translation*
- *The mitoribosome:LRPPRC-SLIRP complex is specific to Metazoa*

Thus, the manuscript has been expanded to a full article format, and it concludes with one page Discussion to put the findings in the context for the reader. Below is the point-by-point response.

Reviewer #1

The manuscript by Singh et al. describes the results of a structural analysis of a novel complex linking transcription and translation in human mitochondria, using cryo-electron microscopy. Indeed, the authors reveal the first structure of the human mitochondrial ribosome (mitoribosome) in complex with LRPPRC (leucine-rich pentatricopeptide repeat-containing protein) and SLIRP (SRA stem-loop-interacting RNA-binding protein), which are both proteins implicated in the interaction with mt-mRNA. In their structure, the authors

show how the mt-mRNA is delivered after its transcription to the mitoribosome that remains attached to the inner mt-membrane.

Although brief, the manuscript is well written and the figures are crystal-clear. The refined structures appear to be of quality based on their statistics and references are sufficient.

The presented structure will be of great interest to the communities of translation regulation, mitochondrial gene expression and the ribosome more broadly. The thorough cryo-EM analysis is also interesting and similar data-processing workflow can be applied to other complexes. I warmly recommend its publication but would however have few suggestions and minor comments:

The authors assign the additional densities to LRPPRC and SLIRP. However, it is not mentioned anywhere in the manuscript how the authors came to this assignment. For instance, mass spectrometry analysis could shed light on the complex composition. Alternatively, the resolution of the map could also be sufficient for accurate assignment when sidechains are clearly resolved, can the authors show some representative portions that allowed them to interpret their structure? Finally, AlphaFold models and their cross-correlation to the map could also suffice, but perhaps the authors need to include some additional supplementary figures/panels to demonstrate their assignment.

Indeed, we used the tools mentioned by the Reviewer, and now added a corresponding description on lines 90-99. In addition, the new Extended Data Fig 1e shows four panels with map regions used for the assignment, and Extended Data Fig 2 shows AlphaFold model with a TLSMD group analysis to demonstrate the assignment, as well as the C-terminal domain that hasn't been modelled.

The authors describe the mRNA densities in which some 42 residues were modeled. However, nowhere in the manuscript they show this very crucial detail on their structure! In my opinion, this is a very important point that merits some more demonstration and highlight. For example, the authors could show fitting of a part of this mRNA portion in its densities, also surface representation of the segmented densities at this region, so the reader can appreciate the resolution and the reliability of this part of the structure.

We now added a presentation of the segmented map in Figure 2a showing a continuous density for mRNA, as well as a panel with map-model overlay in the LRPPRC region. We also toned down the statements regarding mRNA, and the revised text is on lines 109-111: "SLIRP is connected to an elongated density on the LRPPRC surface that is also associated with six of the mitoribosomal proteins and corresponds to the endogenous mRNA".

The authors refer to an Extended Data Figure 5c, which doesn't exist (Page 3 line 86).

Fixed.

Extended Data Figure 2c requires some explanation, but perhaps it is not the most efficient way to represent the complex interactions at the SSU beak.

We made a new Fig. 2b) showing the complete schematics of protein-protein interactions in the complex as connected nodes, which is consistent with the protein organization in the model, and the density segmented map is shown side by side.

Reviewer #2

Overview

Singh et al. report a structural model of the mitochondrial ribosome bound to mRNAs it is translating and the regulatory proteins LRPPRC-SLIRP associated with the mRNA entrance channel. This structure is of great importance for both the insight it offers into fundamental gene expression mechanisms and its relevance to human disease. The use of a native purification scheme from mitochondria provides strong support that the presented architecture is of physiological relevance, and this is a technical achievement to be applauded.

There are no flaws that prohibit publication, and predictions made are fully testable. The work has impact beyond the immediate field of mitochondrial translation: parallels are drawn to molecular mechanisms of bacterial gene expression recently presented that will attract broad interest. The methodology is cutting-edge and the work will be appreciated by many interested in structure determination by cryo-EM. The length of text appears appropriate; as detailed below, an increase in waiting towards interpretation of the results and away from describing the model is needed in this reviewer's opinion.

Major concern

The authors provide all the data and figures that are typically essential for a reader to evaluate the quality of a structural model. However, as noted in the manuscript, the region of this complex of greatest interest (LRPPRC, SLIRP, mS31, mS39 and protruding mRNA) is dynamic. The depth of structural information available for each of these components should be presented more clearly as the conclusions that are drawn from the structure require an indication of their degree of confidence. This concern is deemed 'major' as this clarification would substantially improve the ability of the field to appreciate which questions have been addressed comprehensively here, and what remains to be further characterised.

Below are some specific suggestions this reviewer believes would improve the manuscript in this regard:

- The focus of the manuscript are the structural features adjacent to the mRNA entrance channel, which consists of LRPPRC, SLIRP, mS31 and mS39. The local resolution of this region is reported as 3.4 Å, but there is significant variation in this region as indicated in ExFig1a. To clarify which features are well-resolved and which are not, the authors should consider (1) presenting a map-model overlay in Figure 1 (rather than model alone), (2)

labelling the local resolution map in ExFig1a with which proteins correspond to which density. This is also important as it avoids the apparent discrepancy that the representative density shown ExFig1c does not appear to $<3.5 \text{ \AA}$ (to this reviewer's eyes). Although the presentation of local B-factor in Fig1 conveys some of this information, it does not make it clear what regions can be interpreted at level of amino acid contacts or specific nucleotides, and the manuscripts conclusions rely on this.

As requested, we now added a presentation of the map-model overlay, which can be found in Fig 2a. It shows the overall map segmented according to the elements discussed in the paper, and four zoom-in panels: a magnified view of the mS39-LRPPRC region with a map-model overlay; a magnified view of the mRNA entrance channel with a map-model overlay; a magnified view of SLIRP with a map-model overlay in two different orientations. We also labeled the local resolution map in the Extended Data Fig 1b with proteins LRPPRC, SLIRP, mS31, mS39 and mRNA and their corresponding densities.

- The model of mRNA in the structure is presented with apparent confidence: 42 residues modelled and residues 31-32 recognised by specific residues of LRPPRC. Presenting a map-model overlay of the mRNA chain alone would indicate to readers the confidence in this aspect of the model. For some fraction of the mRNA chain it is likely the resolution prevents unambiguous modelling; given the capacity for ssRNA to expand or contract, could the authors instead indicate a range of RNA lengths that may be sequestered by the channel of proteins and describe the model as 'representative'? This conclusion is particularly important given a primary role of LRPPRC-SLIRP is in protecting/unwinding RNA: the field would benefit from knowing the best estimate of how many nucleotides are protected by the proteins.

We agree with the Reviewer that the indication of the confidence in the mRNA aspect of the model was overestimated. Therefore, we removed the statement regarding the residues, and added panels describing the data as described above.

In addition, to further clarify the role of LRPPRC, we performed RNAseq analysis shown in Fig 3b and metabolic labeling with [^{35}S]-labeled methionine of newly-synthesized mitochondrial polypeptides shown in Fig 3c. It revealed that in the LRPPRC-knockout, transcripts from the heavy strand were lowered by 1.5-4-fold, while the single light strand-encoded ND6 mRNA was not affected as reported, and the effect of the LSFC mutation on RNA stability was limited to six transcripts. Metabolic labeling indicated that incorporation of the radiolabeled amino acid into most newly synthesized mitochondrial proteins is severely decreased in LRPPRC-KO cells. However, there were differential effects among transcripts. Finally, we performed mitoribosome profiling in Fig 4 and overall showed a decreased translation efficiency in LRPPRC-knockout cells for COX1 and COX2 transcripts and the bicistronic ATP8/ATP6 transcript. On the contrary, in LSFC cells, TE was not decreased for any transcript and even increased for several, particularly COX3 and NDI1, COX1, and CYB. These new data is described on page 9.

To support the role of LRPPRC in mRNA binding, we determined the average length of the mitoribosome-protected fragments using mitoribosome profiling (Fig 4b, and page 12). In the

LRPPRC-knockout cells, we observed a decrease in the average protected fragment length. The average protected fragment length in LSFC cells was similar to the LRPPRC-KO, suggesting that whereas the mutant protein participates in translation, it does so differently than the wild-type protein.

The focus of this manuscript, as indicated in the title, is on a translation activation mechanism. However, while clear detail is provided in describing the structural model, very little text is devoted to explaining this mechanism.

We hope the reviewer finds the revised version with additional experimental data from RNAseq and mitoribosome profiling more insightful than the original structural description. To better reflect the overall content of the study, we changed the title to “Structural basis of LRPPRC-SLIRP-dependent translation by mitoribosome”.

Could the authors elaborate on the connection between mRNA ‘hand-off’ and translation activation:

- Is the unfolding of mRNA secondary structure the primary activating role of LRPPRC, as was proposed for the NusG in the coupled E. coli expressome (PMID: 32820062)? The E. coli ribosome has intrinsic RNA helicase activity that has been localised to specific residues (PMID: 15652481) – are these conserved in the human mitoribosome, and if not perhaps LRPPRC-SLIRP provides essential helicase activity?

To address this point, we performed a helicase analysis, as requested the Reviewer. The summary of the results is described on lines 156-163 of the revised manuscript with the suggested references. Helicase motifs have been individually checked against the LRPPRC sequence, however no conserved helicase motifs have been found. Manual inspection did indicate a potential weak conservation, however, none of the regions occurs in the modeled part.

In Discussion, lines 320-324 - we added “LRPPRC does not have a helicase activity or an allosteric mechanism, but rather acts as a docking platform for mRNA and SLIRP to the mitoribosome. The docking of LRPPRC is realized through the mitoribosomal proteins mS39 and the N-terminus of mS31, that together recognize eight of the LRPPRC helical repeats”.

- It is not clear whether the authors propose the LRPPRC-SLIRP complex is loaded onto the ribosome with the mRNA or if a complex with the ribosome pre-exists arrival of the mRNA. It is indicated to bind co-transcriptionally (line 36) but, in contrast, a suggestion is made that it is part of SSU biogenesis (line 146). If the latter is true, LRPPRC-SLIRP could be viewed as a recruitment factor that increases the affinity of mitoribosomes for mRNA (line 71), and this represents a distinct translation activation mechanism. This could be highlighted. A conclusion on the mechanistic model should be included in the Abstract.

On pages 9-10, we now address the question whether LRPPRC-SLIRP preexist as a recruitment factor in a complex with the mitoribosome prior to the arrival of mitochondrial mRNAs for increased affinity, or alternatively, they first bind mRNAs to load them onto the mitoribosome. We generated an LRPPRC-knockout cell line that was rescued with either a

wild-type LRPPRC or a variant carrying the LSFC founder mutation A354V (Extended Data Fig. 4). The steady-state levels of the LSFC variant were reduced by 60%, suggesting protein instability as reported in patients, and the levels of SLIRP were equally decreased. As mentioned above, we then implemented an RNAseq approach and measured translation efficiency (Fig. 4a).

The newly added data suggest that LRPPRC-SLIRP is not a universal preexisting mitoribosomal element with high mRNA affinity but is instead required for the translation of specific transcripts. This is consistent with our two structural observations: partial occupancy and relative stability of LRPPRC on the mitoribosome. The partial occupancy on the mitoribosome, evidenced by the relatively weak cryo-EM density (Extended Data Fig. 1b), and the stability is evidenced by the B-factor range for this region that is similar to integral mitoribosomal components, such as L1 and L7/L12 stalk (Fig. 1a). This summary appears on page 10 of the revised manuscript.

As requested, we included in the Abstract “Taken together, our data suggest that LRPPRC-SLIRP does not preexist on the mitoribosome as its structural element but rather acts in recruitment of specific mRNAs to modulate their translation.”

Minor concerns

The following list contains suggestions that this reviewer believe would significantly aid the clarity of the manuscript:

- Line 19: how does this complex ‘act as a carrier of genetic information’ in a way that it is different to how mRNA alone does. Perhaps the authors mean the complex provides efficiency in gene expression?

Removed.

- Lines 29-32: The wording of the opening sentences implies LRPPRC exists in mitochondria because it provides mRNA protection that in bacteria is provided by transcription-translation coupling (TTC). If this is intentional it is not a correct comparison: most translation in bacteria occurs on mRNA that has been released from RNAP and is not within the nucleoid; further, bacteria contain other proteins that confer mRNA stability. Could the authors take a different approach to the introduction?

We reworded the mentioned paragraph and removed the logical chain. The revised text describes only scientific data without implying mRNA protection. Lines 52-61: “In Escherichia coli, a functional transcription-translation coupling mechanism has been characterised involving a physical association of the RNA polymerase with the SSU, termed the expressome⁶⁻⁸. In mammalian mitochondria, nucleoids are not compartmented with protein synthesis; mitoribosomes are independently tethered to the membrane^{9,10}, and no coupling with the RNA polymerase has been reported. The 130-kDa protein factor LRPPRC (leucine-rich pentatricopeptide repeat-containing protein), a member of the Metazoa-specific

pentatricopeptide repeat family, was reported to act as a global mitochondrial mRNA chaperone that binds co-transcriptionally¹¹⁻¹⁴. LRPPRC is an integral part of the post-transcriptional processing machinery required for mRNA stability, polyadenylation, and translation”.

- Lines 36-37: could this sentence be clarified: LRPPRC both stabilizes and destabilizes mRNA secondary structures? Is it possible the authors mean to remove the word ‘structure’ in line 37? The term ‘global RNA chaperone’ used by Siira et al. may also be useful.

We removed the conflicting text and added, and the revised version reads (lines 58-59): “... a global mitochondrial mRNA chaperone that binds co-transcriptionally”.

- Line 45: missing full stop.

Added.

- Line 47: the phrasing ‘might’ undersells the known cooperation between LRPPRC and SLIRP. The interaction of LRPPRC and SLIRP has been studied in some detail in reference [20] and this could be indicated here.

We removed “might” and added a citation to the suggested references (line 71): “The interaction of LRPPRC and SLIRP in vitro has been previously studied”.

- Line 54: should state ‘entry channel’ to distinguish from mRNA exit channel.

Added.

- Line 56: local resolution of which specific region should be indicated in these brackets.

Added on line 91.

- Line 56: ‘trapped’ should perhaps be changed to ‘visualised’ or similar: no method of trapping the complex was employed other than the freezing process.

Removed.

- Line 58: could the authors indicate the exact residue numbers included in the modelled ‘N-terminal domain’ and unmodelled ‘C-terminal domain’.

Residue numbers have been added for both.

- Line 61: ‘outspreads’ could be changed to a more common word: e.g. ‘projects’ or ‘protrudes’.

Changed.

- Line 62: the use of terms ‘NES’ vs ‘NES-rich’ should be consistent with Fig. 1. Can the

authors confirm it is necessary to introduce the terms ‘head’ and ‘tail’ when it appears these domains have names already?

Changed in the revised Fig 3a.

- Fig.1c: could this include a domain architecture diagram making it clear that a C-terminal domain is present in LRPPRC that was not modelled?

Added.

- Line 63: ‘completion’ rather than completing.

It has been removed now.

- Line 64: the manuscript would benefit from a justification of why this density can be confidently assigned to SLIRP. Was mass spectrometry analysis performed to confirm SLIRP is present in reasonable abundance? Was the interaction interface previously determined (ref [20]) used to assign the density? Does AlphaFold predict a LRPPRC-SLIRP interaction consistent with this?

We added on lines 107-108: “Consistent with mass spectrometry analysis²³ and the interaction interface previously determined³⁰, the remaining associated density was assigned as SLIRP”.

In addition, we performed TLS analysis of the LRPPRC model, where we defined the C-terminal domains as individual segments, and it indicated potential flexibility between domains (Extended Data Fig. 2). The model was divided into TLS segments (N), and single chain TLSMD is performed on all atoms using the isotropic analysis model. Instead of using atomic B-factors, the values for a per residue confidence score of AlphaFold called predicted local distance difference test (pLDDT) were used as reference to calculate the least squared residuals against the corresponding values calculated by TLSMD analysis. This is based on the assumption that local mobility of the model should be inversely correlated with the pLDDT score. AlphaFold pLDDT values and the corresponding calculated values were plotted for every iteration to monitor improvement in prediction and across the length of LRPPRC. The data in Extended Data Fig. 2 is presented for N=4, where segments 1 and 2 (residues 60-373 and 374-649) correspond to the modeled region, whereas segments 3 and 4 correspond to the remaining domains that could not be modeled.

- Line 69: from the start of the E-site?

Removed.

- Line 74: crosslinking mass spectrometry data and mutational analysis

Added.

- Line 76: for clarity: ‘contributes to a corridor for the mRNA that extends that formed by mS31 and mS39’ or similar.

Changed.

- Line 80: it should be stated here or elsewhere that SLIRP is an RRM-containing protein. Could the authors comment on whether the interface of SLIRP that contacts mRNA parallels other known RRM-RNA interactions? Were specific basic or aromatic residues identified that are predicted to be essential for RNA-binding activity?

We now added the analysis in Fig. 2c that shows the arrangement of RNP1 and RNP2 with respect to the mRNA which is similar to that observed in previously reported structures of other RRM proteins. The figure shows RRM containing proteins: SLIRP, hnRNPA1/B2 (PDBID 5WWE), PolyA binding protein (PABP, PDBID 1CVJ) and Nucleolin (PDBID 1RKJ) in complex with RNA. A complementary sequence alignment is shown in Extended Data Fig. 3.

- Line 86: no ExFig5 was provided.

Corrected.

- Line 95: should be: hand-off of the mRNA. The term ‘handing-off’ is used throughout the manuscript, but are ‘delivery’ or ‘transfer’ more common terms for this idea? If not, this should be ‘hand-off’ for a noun.

Changed.

- Line 98: the residue numbers of the arginines should be stated. A figure for this would be useful.

We removed it, because the quality of the density is insufficient.

- Line 100: ‘analysis of recombinant LRPPRC’ would be more specific.

Changed.

- Line 106: ‘contact-2’ has not been introduced before this line (nor contact-1).

We added a paragraph describing the contact sites on lines 255-263.

- Line 120: metazoa should be capitalized

Fixed.

- Line 126: ‘therefore’ rather than ‘therewith’

Removed.

- Line 130: the mRNA entrance channel in bacteria is primarily defined by proteins S3, S4 and S5. The mentioned rRNA regions border these. Could the authors comment on whether S3,S4 and S5 are conserved in their structural roles within the mitoribosome, and if they are why is the loss of the rRNA is worth noting here?

We performed the analysis and described the results on lines 156-163: “Since in E. coli, the expressome-mediating protein NusG was proposed to regulate mRNA unwinding¹, and SSU proteins uS3 and uS4 have an intrinsic RNA helicase activity²⁵, we next analysed if LRPPRC and the corresponding region in the mitoribosome might have similar functions. Particularly, we performed a search for known helicase signature motifs²⁶ in the LRPPRC sequence. The alignments have not detected conserved helicase sequences, and therefore, no phylogenetic relationship can be drawn to RNA helicases. In the mitoribosome, where the mRNA channel entry site is located, a bacteria-like ring-shaped entrance is missing, the entrance itself has shifted, and its diameter expanded¹⁹. We conclude that the LRPPRC is not an mRNA helicase, and the entry to the mitoribosomal channel does not play a role in disruption of the mRNA secondary structure.”

- Line 132: The chosen expressome structure for comparison is not the best: the mRNA path was clearly resolved only in the structure with a shorter mRNA (PDB: 6ZTJ). Here, the mRNA travels across the surface of ribosomal protein S3 (see PMID: 32820062 figure 2). Do the authors suggest the path is comparable in the involvement of S3? It appears the similarity to the expressome is more general: a positively charged surface on the exterior of the ribosome immediately adjacent to the mRNA entrance channel. The authors should also cite a similar channel mechanism formed by NusA in the M. pneumoniae expressome (PMID: 32732422 Fig. 3).

We reanalysed the data with the PDB 6ZTJ, as well as the M. pneumoniae structure. The corresponding panels in Fig. 5 have been updated. On lines 296-297, we added: “Similar conclusion is reached from comparison with the M. pneumoniae expressome.”

- Line 139: ‘mito-specific’ should be mitoribosome-specific.

Changed.

- Line 141: a mechanisms of specific mechanisms. This could be described as a convergent evolution of translation mechanisms with reference to bacterial transcription-translation coupling factors.

- Line 143: ‘the’ central regulator of gene expression sounds like an overstatement: ‘a’ central regulator?

Changed and removed “central”.

- Line 146: citations appear to be the wrong order: 27-26.

Fixed.

- Line 148-149: the wording of this requires revision to be clear what is meant.

Fixed.

- Could the positions of mutations in LRPPRC/SLIRP known to be disease-related be mapped on the structural model in an Extended data figure?

Mapped in Fig. 3a and Extended Data Fig. 2a.

- Reference 6 should be 2012: could the authors check accuracy of all references.

Changed and checked.

Decision Letter, first revision:**Message:** 17th Feb 2023

Dear Dr. Amunts,

Thank you again for submitting your manuscript "Structural basis of LRPPRC-SLIRP-dependent translation by the mitoribosome". I sincerely apologize for the delay in responding, which resulted from the difficulty in obtaining suitable referee reports. Nevertheless, we now have comments (below) from the 2 original reviewers who evaluated your manuscript as well as a 3rd reviewer who was recruited to evaluate the additional sequencing data added in the revised manuscript. In light of those reports, we remain interested in your study and would like to see your response to the comments of the referees, in the form of a revised manuscript.

You will see that while Reviewers #1 and #2 now support the publication of this work, Reviewer #3 has major concerns about the analysis and interpretation of the sequencing results. This reviewer provides guidance on how to improve this part. However, please reach out to me if you wish to discuss alternative approaches to the revision as, editorially, we understand that while extensive revision of the sequencing part will strengthen the manuscript, it may be outside the scope of the present work. Otherwise, please be sure to address/respond to all concerns of the referees in full in a point-by-point response and highlight all changes in the revised manuscript text file. If you have comments that are intended for editors only, please include those in a separate cover letter.

We expect to see your revised manuscript within 6 weeks. If you cannot send it within this time, please contact us to discuss an extension; we would still consider your revision, provided that no similar work has been accepted for publication at NSMB or published elsewhere.

Reporting Summary:

When submitting the revised version of your manuscript, please pay close attention to our [href="https://www.nature.com/nature-portfolio/editorial-policies/image-integrity">Digital Image Integrity Guidelines](https://www.nature.com/nature-portfolio/editorial-policies/image-integrity). and to the following points below:

Please note that all key data shown in the main figures as cropped gels or blots should be presented in uncropped form, with molecular weight markers. These data can be aggregated into a single supplementary figure item. While these data can be displayed in a relatively informal style, they must refer back to the relevant figures. These data should be submitted with the final revision, as source data, prior to acceptance, but you may want to start putting it together at this point.

Data availability: this journal strongly supports public availability of data. All data used in accepted papers should be available via a public data repository, or alternatively, as Supplementary Information. If data can only be shared on request, please explain why in your Data Availability Statement, and also in the correspondence with your editor. Please note that for some data types, deposition in a public repository is mandatory - more information on our data deposition policies and available repositories can be found below: <https://www.nature.com/nature-research/editorial-policies/reporting-standards#availability-of-data>

We require deposition of coordinates (and, in the case of crystal structures, structure factors) into the Protein Data Bank with the designation of immediate release upon publication (HPUB). Electron microscopy-derived density maps and coordinate data must be deposited in EMDb and released upon publication. Deposition and immediate release of

NMR chemical shift assignments are highly encouraged. Deposition of deep sequencing and microarray data is mandatory, and the datasets must be released prior to or upon publication. To avoid delays in publication, dataset accession numbers must be supplied with the final accepted manuscript and appropriate release dates must be indicated at the galley proof stage.

[Redacted]

Sincerely,
Sara

Sara Osman, Ph.D.
Associate Editor
Nature Structural & Molecular Biology

Reviewers' Comments:

Reviewer #1:

Remarks to the Author:

The authors have addressed all of the reviewer's raised points. The manuscript looks quite fit for publication in my opinion.

Reviewer #2:

Remarks to the Author:

The authors have addressed all major and minor concerns raised by the reviewers.

The inclusion of new data significantly extends the conclusions that can be drawn from this work. In particular, a major outstanding question has been addressed: whether the LRPPRC-SLIRP complex is a general activator of translation that acts on all mitochondrial mRNAs, or a specific activator for a subset of transcripts. This question is closely linked to that of the chronology of steps: whether LRPPRC-SLIRP pre-exists on mitochondrial ribosomes ready to act on any incoming mRNA, or arrives with specific transcripts. The presented evidence robustly supports the latter conclusion in each case. The new data also examines the molecular consequences of disease-linked mutation, which is expected to significantly increase the impact of the study.

I strongly recommend publication and am grateful to have had the opportunity to provide feedback on this research.

Reviewer #3:

Remarks to the Author:

This manuscript by Singh et al. reports a cryo-EM study of ribosomes isolated from human mitochondria, revealing the structure of two important but poorly understood proteins, LRPPRC and SLIRP, in complex with the ribosome. These proteins associate with the mRNA near the site where it enters the ribosome, an overall architecture that is strikingly different from the mRNA entry channel in bacterial or human cytosolic ribosomes. Given that LRPPRC and SLIRP have been previously shown to bind and stabilize mt-mRNAs, this work may have important implications for the mechanism of translational initiation in mitochondria. This is clearly an impressive result from a technical point of view, and I have no real concerns about the beautiful structure that they report and their descriptions of it.

The biological implications of the structural findings reported here, however, are not so clear. LRPPRC and SLIRP may be part of a mechanism to recruit mRNA to the mt-ribosome. It is also logically possible that they associate with mRNA the entire time it is bound to the mt-ribosome, or that they are constitutive components of the ribosome. The authors report that only a fraction of particles have these proteins bound, but this is only indirect evidence against them being a constitutive part of the ribosome. Perhaps they fall off easily during the purification.

In line 183, the authors ask this exact question, is LRPPRC in the mt-ribosome always, or does it first bind mRNAs? But then they report an RNAseq experiment that doesn't really address that specific question. Even if LRPPRC bound to the ribosome constitutively, it could still bind selectively to some mRNAs and not others. It would be better to frame the RNAseq experiment in terms of "does LRPPRC deliver all mRNAs to the mt-ribosome or is it selective."

Looking at the RNAseq data in the knockout (Fig 3b, left), where levels of all but one mt mRNA go down 2-4 fold, I would not argue that LRPPRC is specific for some mRNAs over others. Furthermore, in the LFSC mutant, all the mt mRNAs except Cox1 appear in the dense cloud of grey dots from all the other nuclear-encoded genes. It doesn't make sense in this case either to argue for selective effects.

On the other hand, there are selective effects in the 35S experiment in Fig 3C. For

example, ND3 and ND4L do not change much on LRPPRC knockdown, whereas Cox2 and Cox3 are nearly undetectable. It would be wonderful if the authors could reconcile these data with the RNAseq and ribo-seq data together. Perhaps the more efficient translation of ND4L and ND3 keeps their expression high even though the transcript levels drop, whereas inhibited translation for Cox2 and Cox1 further limits their expression.

In any case, it is difficult to make a compelling argument for a role in initiation when knocking out LRPPRC may have a direct effect on destabilizing mRNAs with or without translation or interaction with the ribosome.

In summary, although I see some evidence of mRNA selectivity in the 35S labeling experiment, the authors have not made a compelling case for mRNA selectivity. They have not proven that LRPPRC is involved in initiation. There are problematic claims regarding the helicase activity of the mt-ribosome and LRPPRC (see below). Given these concerns, the biological implications of this lovely structural study are not fully developed, and some revision and rethinking are required prior to publication.

Additional comments:

Line 54: In contrast, in mammalian mitochondria...

I would appreciate more background in the introduction about how are mRNAs recruited into the mt-ribosome. I assume there are no 5'-caps like in cytosolic eukaryotic mRNAs or Shine-Dalgarno sequences like in bacteria. But what do we know about initiation in mitochondria?

Line 57: new paragraph: The 130-kD protein factor LRPPRC...

Line 73: "We previously used mass-spectrometry analysis of natively purified mitoribosomes to detect the presence of LRPPRC-SLIRP that is correlated with a density for mRNA on the mitoribosome" The first part of this sentence makes sense. I don't understand how it relates to the second part, however. How does a crosslink correlate with density?

Line 93-94: "Nine of which form a superhelical-ring structure" – please give the numbers of the helices so that this observation can be mapped onto the structure in Fig 1A.

Larger protein labels in the Fig 1A detail would help the reader (i.e. bigger font for mS31, etc).

Fig 2A – I can't make much sense of the structures in the boxes. Are they cited in the text? What are they trying to show?

The site of mRNA entry in Fig 2A in the SSU is very different from ring of uS3, uS4, uS5 in other ribosomes. The mRNA extends all the way into the head/beak of the SSU. I would like to see this aspect of the structural work developed more in the text.

Line 156-163: The discussion of the mRNA entry site is couched in terms of whether it has helicase activity, which I think is a mistake. I would like to see the entry channel discussed in from a structural point of view (not a biochemical point of view). The authors don't show whether mt-ribosomes or these factors have helicase activity. Lines 162-163 I

particularly object to “We conclude that LRPPRC is not an mRNA helicase, and the entry to the mitoribosomal channel does not play a role in disrupting the mRNA secondary structure.” Of course, it may be that LRPPRC by acting as an mRNA chaperone melts secondary structure without hydrolyzing ATP and scanning down the mRNA like a helicase does.

Fig 3A belongs in Fig 1 and not with the expression data in Fig 3.

Fig 4A: ND6 RNA is down two-fold but they said above it was unaffected. There is no ND6 seen in the 35S Met experiment. But here they argue that the translation of ND6 is 3-4 fold higher in the knockout? It just doesn't add up.

Fig 4A: TE vs RNAseq seems an overly complicated way of plotting these data. The RNA levels appear in both the y and x axes. Do they really want to talk about TE as a function of RNA level? I think they really want is to plot translation (ribo-seq) a function of mRNA level (rna-seq) and then discuss changes in translation, and TE.

Fig 4A, right. What does it mean that the TE values go up in the LSFC mutant? I don't think these data are very helpful. I can't make any real sense of them.

Fig 4A: Since they have replicates for the ribo-seq, the authors could use a program like xtail to generate a volcano plot for TE giving the significance of the translational changes that they see. (xtail is like DEseq3 but it works for ribo-seq).

Lines 210-216. The observation of partial occupancy of LRPPRC doesn't necessarily support the claim that it is selective for some mRNAs. The partial occupancy may simply arise from the fact that it is only present on some sub-population of particles, perhaps during initiation but not during elongation.

Line 227. Why not put this description of the structure together with the helical interactions diagram above in or after Fig 1. It seems strange to come back to it here.

Line 240-241 How does the nucleotide numbering work? What is the distance from the first nt in the P site codon to nucleotide 31/32?

Fig 4b: Why would ribosome footprints be SMALLER in the LRPPRC knockout, when it should protect the mRNA from degradation at the 3'-end of the footprint? I wonder if they could gain any insight from looking at the footprint sizes at start codons vs internal ORF footprints. Also, are there differences in length at the 5' end or the 3' end of the footprints?

Author Rebuttal, first revision:

We thank Reviewers #1 and #2 for taking the time to go through the manuscript again. We thank Reviewer #3 for providing additional constructive suggestions on how to improve the study and its presentation. In the revised version, we addressed all the requests and followed the valuable suggestions. Below is the point-by-point response to Reviewer #3.

Reviewer #1:

Remarks to the Author:

The authors have addressed all of the reviewer's raised points. The manuscript looks quite fit for publication in my opinion.

Reviewer #2:

Remarks to the Author:

The authors have addressed all major and minor concerns raised by the reviewers.

The inclusion of new data significantly extends the conclusions that can be drawn from this work. In particular, a major outstanding question has been addressed: whether the LRPPRC-SLIRP complex is a general activator of translation that acts on all mitochondrial mRNAs, or a specific activator for a subset of transcripts. This question is closely linked to that of the chronology of steps: whether LRPPRC-SLIRP pre-exists on mitochondrial ribosomes ready to act on any incoming mRNA, or arrives with specific transcripts. The presented evidence robustly supports the latter conclusion in each case. The new data also examines the molecular consequences of disease-linked mutation, which is expected to significantly increase the impact of the study.

I strongly recommend publication and am grateful to have had the opportunity to provide feedback on this research.

Reviewer #3:

Remarks to the Author:

This manuscript by Singh et al. reports a cryo-EM study of ribosomes isolated from human mitochondria, revealing the structure of two important but poorly understood proteins, LRPPRC and SLIRP, in complex with the ribosome. These proteins associate with the mRNA near the site where it enters the ribosome, an overall architecture that is strikingly different from the mRNA entry channel in bacterial or human cytosolic ribosomes. Given that LRPPRC and SLIRP have been previously shown to bind and stabilize mt-mRNAs, this work may have important implications for the mechanism of translational initiation in mitochondria. This is clearly an impressive result from a technical point of view, and I have no real concerns about the beautiful structure that they report and their descriptions of it.

The biological implications of the structural findings reported here, however, are not so clear. LRPPRC and SLIRP may be part of a mechanism to recruit mRNA to the mt-ribosome. It is

also logically possible that they associate with mRNA the entire time it is bound to the mt-ribosome, or that they are constitutive components of the ribosome. The authors report that only a fraction of particles have these proteins bound, but this is only indirect evidence against them being a constitutive part of the ribosome. Perhaps they fall off easily during the purification.

In line 183, the authors ask this exact question, is LRPPRC in the mt-ribosome always, or does it first bind mRNAs? But then they report an RNAseq experiment that doesn't really address that specific question. Even if LRPPRC bound to the ribosome constitutively, it could still bind selectively to some mRNAs and not others. It would be better to frame the RNAseq experiment in terms of "does LRPPRC deliver all mRNAs to the mt-ribosome or is it selective."

Looking at the RNAseq data in the knockout (Fig 3b, left), where levels of all but one mt mRNA go down 2-4 fold, I would not argue that LRPPRC is specific for some mRNAs over others. Furthermore, in the LFSC mutant, all the mt mRNAs except Cox1 appear in the dense cloud of grey dots from all the other nuclear-encoded genes. It doesn't make sense in this case either to argue for selective effects.

On the other hand, there are selective effects in the 35S experiment in Fig 3C. For example, ND3 and ND4L do not change much on LRPPRC knockdown, whereas Cox2 and Cox3 are nearly undetectable. It would be wonderful if the authors could reconcile these data with the RNAseq and ribo-seq data together. Perhaps the more efficient translation of ND4L and ND3 keeps their expression high even though the transcript levels drop, whereas inhibited translation for Cox2 and Cox1 further limits their expression.

In any case, it is difficult to make a compelling argument for a role in initiation when knocking out LRPPRC may have a direct effect on destabilizing mRNAs with or without translation or interaction with the ribosome.

In summary, although I see some evidence of mRNA selectivity in the 35S labeling experiment, the authors have not made a compelling case for mRNA selectivity. They have not proven that LRPPRC is involved in initiation. There are problematic claims regarding the helicase activity of the mt-ribosome and LRPPRC (see below). Given these concerns, the biological implications of this lovely structural study are not fully developed, and some revision and rethinking are required prior to publication.

We thank the Reviewer for the constructive criticism. We did our best to follow all the suggestion in order to improve the manuscript.

The RNAseq experiment can only speak about RNA steady-state levels, and not directly about whether the mRNAs are delivered to the mitoribosome. To clarify our point, we added on line 225: "... indicating LRPPRC's role in heavy strand mRNA stability is non-specific."

We agree with the reviewer that LRPPRC stabilizes and protects all H-strand mRNAs, although ND3 is largely protected in the absence of LRPPRC, consistently with the ND3 protein synthesis data (Fig 3b). The specificity that we are claiming regards the LRPPRC role in handing off transcripts for translation.

Regarding the mitochondrial protein synthesis experiment in Fig 3b, we had unfortunately made an error in the labeling of ND3, ND4L, ND6, which we have now corrected. We have performed WB analyses, included below, which allowed us to identify each band. Now, we can clearly see that the translation of ND3 is slightly affected, and ND6 is not affected in the absence of LRPPRC. We have amended the figure and quantified the translation of ND3.

We have amended the text on lines 240-242 to reflect the congruency between the data from ribosome profiling and 35S- labeling data, as requested: “*Thus, our data suggest that LRPPRC-SLIRP is required for the translation of some transcripts, in agreement with the metabolic labeling of newly synthesized mitochondrial products (Fig. 3b).*”

Regarding ‘translation efficiency’, we now added a clarification on lines 235-238: “*These data show a decreased TE, calculated by dividing spike-in normalized ribosome footprint reads by the spike-in normalized RNA sequencing reads (in other words, how well a particular transcript is translated)*”.

Additional comments:

Line 54: In contrast, in mammalian mitochondria...

Changed.

I would appreciate more background in the introduction about how are mRNAs recruited into the mt-ribosome. I assume there are no 5'-caps like in cytosolic eukaryotic mRNAs or Shine-Dalgarno sequences like in bacteria. But what do we know about initiation in mitochondria?

Great suggestion. We now added this background on lines 55-62 with references to the recent studies from Rodnina and Puglisi labs: *'In addition, human mitochondrial mRNAs and the mitoribosome do not have the Shine–Dalgarno (SD) and anti-SD sequences that are used to recruit mRNA to SSU in bacteria. Mitochondrial mRNAs also lack cap 5' modifications, which is a hallmark of eukaryotic cytosolic mRNAs translation initiation. In the cytosol, mRNA is recruited to a pre-initiation complex, consisting of the SSU and translation initiation factors, which then scans along the 5' untranslated region to find the start codon^{13,14}. No equivalent mechanism has been found in mitochondria, and thus, how mRNAs are delivered for translation in mitochondria remained unknown.'*

Since our structural data is consistent with the published translation initiation complex in mitochondria, this information has now been added on lines 194-195 with a corresponding citation: *'Here, the mRNA is handed to mS31-mS39, consistently with a translation initiation complex⁴⁵'*.

Line 57: new paragraph: The 130-kD protein factor LRPPRC...

Fixed, thank you.

Line 73: "We previously used mass-spectrometry analysis of natively purified mitoribosomes to detect the presence of LRPPRC-SLIRP that is correlated with a density for mRNA on the mitoribosome" The first part of this sentence makes sense. I don't understand how it relates to the second part, however. How does a crosslink correlate with density?

Revised to *'Previous analysis showed a correlation between presence of LRPPRC and mRNA on the mitoribosome'*.

Line 93-94: "Nine of which form a superhelical-ring structure" – please give the numbers of the helices so that this observation can be mapped onto the structure in Fig 1A.

Changed to *'It allowed us to model 34 α -helices, 17 of which (α 2-18) form a ring-like architecture ...'*

Larger protein labels in the Fig 1A detail would help the reader (i.e. bigger font for mS31, etc).

Size of the protein labels has been increased as suggested.

Fig 2A – I can't make much sense of the structures in the boxes. Are they cited in the text? What are they trying to show?

We revised this figure to simplify and clarify the closeup views in the boxes. The legend has also been revised to: *'The model and map for mS39-LRPPRC-SLIRP and corresponding bound mRNA residues are shown in closeup views on the left, and arginines involved in mRNA binding are indicated. The bottom closeup views show SLIRP with its associated densities for LRPPRC and mRNA.'*

The site of mRNA entry in Fig 2A in the SSU is very different from ring of uS3, uS4, uS5 in other ribosomes. The mRNA extends all the way into the head/beak of the SSU. I would like to see this aspect of the structural work developed more in the text.

Line 156-163: The discussion of the mRNA entry site is couched in terms of whether it has helicase activity, which I think is a mistake. I would like to see the entry channel discussed in from a structural point of view (not a biochemical point of view). The authors don't show whether mt-ribosomes or these factors have helicase activity. Lines 162-163 I particularly object to "We conclude that LRPPRC is not an mRNA helicase, and the entry to the mitoribosomal channel does not play a role in disrupting the mRNA secondary structure." Of course, it may be that LRPPRC by acting as an mRNA chaperone melts secondary structure without hydrolyzing ATP and scanning down the mRNA like a helicase does.

As suggested, we removed the misleading sentence and added structural data. The revised paragraph on lines 164-172 reads - *'Since in E. coli, the expressome-mediating protein NusG was proposed to regulate mRNA unwinding⁸, and SSU proteins uS3 and uS4 have an intrinsic RNA helicase activity⁴³, we searched for known helicase signature motifs⁴⁴ in the LRPPRC sequence, but no such motifs were present. In the mitoribosome, where the mRNA channel entry site is located, a bacteria-like ring-shaped entrance is missing, the entrance itself has shifted, and its diameter expanded³⁰. The mRNA extends all the way into the head/beak of the SSU stabilised by mitoribosome-specific components: mS39 helical repeats, mS35 N-terminus that extends from the side of the SSU head, and N-terminal extension of uS9m that contacts the mRNA nucleotide at position 15 (numbered from the E-site).'*

Fig 3A belongs in Fig 1 and not with the expression data in Fig 3.

We agree and moved the panel to Fig 1c.

Fig 4A: ND6 RNA is down two-fold but they said above it was unaffected. There is no ND6 seen in the 35S Met experiment. But here they argue that the translation of ND6 is 3-4 fold higher in the knockout? It just doesn't add up.

The data presented in Fig 4 are normalized by spike-in, whereas the data in Fig 3 are normalized by total cellular RNA, which may explain some differences across experiments. Due to their short sizes, we have found that *ND6* and *ATP8* transcripts show a variability in RNA-seq and ribo-seq libraries. On the other hand, previous reports have clearly established, using Northern blot analyses, that *ND6* mRNA is stable in the absence of LRPPRC (Ruzzenette et al., 2012).

Regarding the newly-synthesized protein, we had unfortunately made an error in the labeling of the 35S-met incorporation assay, which we have now corrected. The ND6 translation is not negatively affected in the absence of LRPPRC.

Fig 4A: TE vs RNAseq seems an overly complicated way of plotting these data. The RNA levels appear in both the y and x axes. Do they really want to talk about TE as a function of RNA level? I think they really want is to plot translation (ribo-seq) a function of mRNA level (rna-seq) and then discuss changes in translation, and TE.

To enhance the clarity of the graph, the Y-axis originally labeled as “translation change (TE), has been changed to the more accurate “Change in translation efficiency (TE)”. We think that plotting ribo-seq on the y axis would be more convoluted because then the RNA abundance element would be present in both dimensions.

Fig 4A, right. What does it mean that the TE values go up in the LSFC mutant? I don't think these data are very helpful. I can't make any real sense of them.

We agree that these data are confusing. We interpret the data to mean that there is a compensatory mechanism that increases translation which remains in the rescue with the LSFC mutant. Nevertheless, we agree with the Reviewer that these data are not helpful, so we have removed the panel from the figure.

Fig 4A: Since they have replicates for the ribo-seq, the authors could use a program like xtail to generate a volcano plot for TE giving the significance of the translational changes that they see. (xtail is like DEseq3 but it works for ribo-seq).

Unfortunately, we cannot use a statistical program like xtail since it requires a complex translome, such as the nuclear-encoded transcriptome, where most transcripts are not experiencing translation regulation, which is not the case here.

Lines 210-216. The observation of partial occupancy of LRPPRC doesn't necessarily support the claim that it is selective for some mRNAs. The partial occupancy may simply arise from the fact that it is only present on some sub-population of particles, perhaps during initiation but not during elongation.

We agree and removed the claim.

Line 227. Why not put this description of the structure together with the helical interactions diagram above in or after Fig 1. It seems strange to come back to it here.

Thank you for this suggestion. We now moved this section up, as suggested. Indeed, the manuscript flows better if all the description of the structure comes together.

Line 240-241 How does the nucleotide numbering work? What is the distance from the first nt in the P site codon to nucleotide 31/32?

The mRNA residues are numbered from the E-site, and we added this information to the text.

Fig 4b: Why would ribosome footprints be SMALLER in the LRPPRC knockout, when it should protect the mRNA from degradation at the 3'-end of the footprint? I wonder if they could gain any insight from looking at the footprint sizes at start codons vs internal ORF footprints. Also, are there differences in length at the 5' end or the 3' end of the footprints?

Perhaps there is a confusion. In the LRPPRC-knockout cells, we observed a decrease in the average protected fragment length, consistently with the structural studies. We now clarified it in the text on lines 243-249: “To support the role of LRPPRC in mRNA binding, we determined an average length of the mitoribosome-protected fragments using mitoribosome profiling (Fig 4b). In the LRPPRC-knockout cells, we observed a decrease in the average protected fragment length, compared to the wild type (Fig 4b). This observation is consistent with the structural data showing the association of LRPPRC with mRNA and the mitoribosome. Previous studies also showed that LRPPRC–SLIRP relaxes structures of mRNAs¹⁶, potentially to expose it to initiate translation¹⁵.”

In addition, we have performed the requested analyses, but they were not informative.

A. Average read size by position in three representative genes. For each position, the average read size in a 10 bp sliding window was calculated and plotted as a single point. The first ~25% of each gene is shown. B. Log2 ratio of the mean of the read sizes at the transcript start (first 2 nt) to the mean of the read sizes from bases 10 to 200. C. Scatter plots of read size vs position for the first ~200 bases of CO1. Black and red points show the read 5' and 3' ends, respectively. Vertical lines are for use in comparing the read ends between the plots. D. Percent of A sites on subcodons 1, 2, or 3 when A sites are calculated from the 3' end of reads (left), or the 5' end of reads (right).

First, we investigated the footprint sizes at start codons vs internal ORF footprints. We tried this a few different ways: by plotting the average size at each position (not shown), by plotting the average size in a sliding window (A), and by calculating the average size at the very 5' end positions (first 2 nt) compared to the average size of the internal positions 10-200 (B). None of these methods showed any consistent differences in the LRPPRC-KO cell line. Next to compare possible 5' vs 3' end extensions we looked at 5' and 3' RPF ends at pause positions where they are visually apparent in V plots (scatter plots of RPF length vs position) (C). This did not work that well because in most cases there are so many fewer reads in the KO that it's hard to compare. Nevertheless, there was no obvious difference on either end. Consistent with this, when A site positions are calculated from RPF 3' ends peaks are generally on the second subcodon position across WT, rescue, and KO cells (D). We redid the A site transformation this time from the 5' end to see if that would result in a difference in A sites in the KO vs rescue. We could find some minor differences but negligible since, in bulk, the subcodon peaks are quite consistent across the cell lines (D).

Decision Letter, second revision:

Message: Our ref: NSMB-BC46505B

1st June 2023

Dear Dr. Amunts,

Thank you for submitting your revised manuscript "Structural basis of LRPPRC-SLIRP-dependent translation by the mitoribosome" (NSMB-BC46505B). It has now been seen one final time by Reviewer #3 and their comments are below. Though the reviewer finds that the paper has improved in revision, the reviewer remains unconvinced of the robustness and interpretability of Figure 4E, and finds that the manuscript is stronger without it. Therefore, we'll be happy in principle to publish the paper in Nature Structural & Molecular Biology, pending revisions to remove this panel, and to satisfy the referees' remaining final requests, as well as to comply with our editorial and formatting guidelines.

We are now performing detailed checks on your paper and will send you a checklist detailing our editorial and formatting requirements in a couple of weeks. Please do not upload the final materials and make any revisions until you receive this additional information from us.

Sincerely,
Sara

Sara Osman, Ph.D.
Associate Editor
Nature Structural & Molecular Biology

Reviewer #3 (Remarks to the Author):

The authors have improved the clarity of the presentation and responded thoughtfully to previous suggestions. Fundamentally I am convinced by their conclusions about the role of LRPPRC. It is clearly required for the stability of several mitochondrial mRNAs. This is clear from the RNAseq experiment. And its loss has profound effects on protein synthesis; these are clear from the 35S studies, for the same set of messages.

But I still struggle with the TE analyses from ribosome profiling. As shown in Fig 4E, when LRPPRC is knocked out, some genes have higher TE and some have lower TE, and the cloud is more or less centered at zero, no change in TE. This is not what you would expect from the deletion of a critical initiation factor that loads mRNA onto ribosomes. You would expect all of the genes to show substantial loss of TE. This means that the messages are binding ribosomes with or without LRPPRC, or more likely, these data are noisy and not telling them what they want to know. Take ND1 and ND2 for example – these proteins are

dramatically reduced in expression in the 35S assay but the ribosome profiling suggests they have higher TE in the LRPPRC knockout. I would argue that this paper is stronger without Fig 4E.

A couple of things about the way the RNAseq data are presented:

Line 224-225: "effect of LSFC mutation on RNA stability was limited to six transcripts." But the six labeled transcripts in colored dots in Fig 3A right are all in the cloud of all the other, cytoplasmic transcripts. Surely these six are not meaningfully different. I wouldn't say there was any effect at all, except on Cox1.

Line 225 "indicating LRPPRC's role in heavy strand mRNA stability is non-specific." The authors added this in response to the previous reviews, and it seems true, but it isn't clear in this context. Better would be : In the LRPPRC-knockout, all but one of the transcripts from the heavy strand were lowered by 1.5-4-fold, suggesting that LRPPRC's role in heavy strand mRNA stability is non-specific.

Finally, the line in the discussion doesn't seem warranted by their data. How can they know if LRPPRC arrived on the mt-ribosome before mRNA or not? How do the profiling data prove this? Line 335 : Furthermore, our mitoribosome profiling data together with the structural analysis show that LRPPRC-SLIRP does not preexist on the mitoribosome as a structural element

Author Rebuttal, second revision:

Reviewer #3:

Remarks to the Author:

The authors have improved the clarity of the presentation and responded thoughtfully to previous suggestions. Fundamentally I am convinced by their conclusions about the role of LRPPRC. It is clearly required for the stability of several mitochondrial mRNAs. This is clear from the RNAseq experiment. And its loss has profound effects on protein synthesis; these are clear from the 35S studies, for the same set of messages.

But I still struggle with the TE analyses from ribosome profiling. As shown in Fig 4E, when LRPPRC is knocked out, some genes have higher TE and some have lower TE, and the cloud is more or less centered at zero, no change in TE. This is not what you would expect from the deletion of a critical initiation factor that loads mRNA onto ribosomes. You would expect all of the genes to show substantial loss of TE. This means that the messages are binding ribosomes with or without LRPPRC, or more likely, these data are noisy and not telling them what they want to know. Take ND1 and ND2 for example – these proteins are dramatically reduced in

expression in the 35S assay but the ribosome profiling suggests they have higher TE in the LRPPRC knockout. I would argue that this paper is stronger without Fig 4E.

We agree. We added a scatter plot showing that ND6 synthesis is not changed, clarified this point in the text on page 10, lines 215-227 and moved the figure to Extended Data Figure 7.

Revised text: ‘To dissect the role of LRPPRC in translation versus RNA stability of each transcript we performed mitoribosome profiling along with matched RNA-seq (Extended Data Fig. 7). Consistent with the metabolic labeling assays and RNA-seq data presented above, in LRPPRC-knockout cells we observe a decrease in inferred protein synthesis and RNA abundance for all mitochondrial transcripts except for ND6 (Extended Data Fig. 7a). The visible correlation between synthesis change and RNA change suggests that much of the effect is a result of the changes in RNA abundance alone. Indeed, when synthesis of each transcript is normalized by its abundance to isolate the effect of translation alone (translation efficiency, TE)^{49,50} we see somewhat less dramatic changes at the translation level (Extended Data Fig. 7b). This analysis highlights the differential effects across transcripts: COX1 and COX2 TE is decreased more than 2-fold in the absence of LRPPRC while ND6 TE is increased more than 2-fold. Thus, our data suggest that LRPPRC-SLIRP has a role in controlling translation efficiency in a transcript-specific manner (Fig. 3b).

A couple of things about the way the RNAseq data are presented:

Line 224-225: “effect of LSFC mutation on RNA stability was limited to six transcripts.” But the six labeled transcripts in colored dots in Fig 3A right are all in the cloud of all the other, cytoplasmic transcripts. Surely these six are not meaningfully different. I wouldn’t say there was any effect at all, except on Cox1.

We agree and changed to ‘... the effect of the LSFC mutation on RNA stability was limited primarily to COX1 transcript.’

Also changed in the abstract accordingly.

Line 225 “indicating LRPPRC’s role in heavy strand mRNA stability is non-specific.” The authors added this in response to the previous reviews, and it seems true, but it isn’t clear in this context. Better would be : In the LRPPRC-knockout, all but one of the transcripts from the heavy strand were lowered by 1.5-4-fold, suggesting that LRPPRC’s role in heavy strand mRNA stability is non-specific.

Changed as suggested.

Finally, the line in the discussion doesn't seem warranted by their data. How can they know if LRPPRC arrived on the mt-ribosome before mRNA or not? How do the profiling data prove this? Line 335 : Furthermore, our mitoribosome profiling data together with the structural analysis show that LRPPRC-SLIRP does not preexist on the mitoribosome as a structural element

Removed.

Final Decision Letter:

Message: 28th Jun 2024

Dear Dr. Amunts,

We are now happy to accept your revised paper "Structural basis of LRPPRC-SLIRP-dependent translation by the mitoribosome" for publication as a Brief Communication in Nature Structural & Molecular Biology.

Your paper will be published online soon after we receive proof corrections and will appear in print in the next available issue. You can find out your date of online publication by contacting the production team shortly after sending your proof corrections.

Please note that *Nature Structural & Molecular Biology* is a Transformative Journal (TJ). Authors may publish their research with us through the traditional subscription access route or make their paper immediately open access through payment of an article-processing charge (APC). Authors will not be required to make a final decision about access to their article until it has been accepted. Find out more about Transformative Journals

Sincerely,
Sara

Sara Osman, Ph.D.
Senior Editor
Nature Structural & Molecular Biology